20

# An important mechanism of regional O<sub>3</sub> transport for summer smog over the Yangtze River Delta in East China

Jun Hu<sup>1</sup>, Yichen Li<sup>2</sup>, Tianliang Zhao<sup>1,\*</sup>, Jane Liu<sup>2, 3</sup>, Xiao-Ming Hu<sup>4</sup>, Duanyang Liu<sup>5,</sup> Yongcheng Jiang<sup>1, 6</sup>, Jianming Xu<sup>7</sup>, Luyu Chang<sup>7</sup>

<sup>1</sup> Collaborative Innovation Center on Forecast and Evaluation of Meteorological Disasters, Key Laboratory for Aerosol-Cloud-Precipitation of China Meteorological Administration, Nanjing University of Information Science & Technology, Nanjing 210044, China

<sup>2</sup> School of Atmospheric Sciences, Nanjing University, Nanjing, 210046, China

<sup>3</sup> University of Toronto, Toronto, M5S 3G3, Canada

10 <sup>4</sup>Center for Analysis and Prediction of Storms, and School of Meteorology, University of Oklahoma, Norman, Oklahoma 73072, USA

<sup>5</sup> Jiangsu Meteorological Observatory, Nanjing, 210008, China

<sup>6</sup>Laboratory of Strait Meteorology, Xiamen Meteorological Observatory, Xiamen Meteorological Bureau, Xiamen 361012, China

15<sup>7</sup> Yangtze River Delta Center for Environmental Meteorology Prediction and Warning, Shanghai 200030, China

Correspondence to: Tianliang Zhao (<u>tlzhao@nuist.edu.cn</u>)

Abstract. Severe ozone (O3) pollution episodes plague a few regions in Eastern China at times, e.g., the Yangtze River Delta (YRD). The formation mechanisms including contributing meteorological factors of these severe pollution events remain elusive. A severe summer smog stretched over the YRD region from August 22 to 25, 2016 with hourly surface O3 concentrations exceeding 300 µg m-3 on August 25 in Nanjing, located in the western YRD. The weather pattern of this episode was characterized by near-surface prevailing easterly wind and continuous high air temperature. The formation mechanism of this O3 episode over the YRD area, particularly the extreme values over western YRD, was investigated using observation data and simulation with the Weather Research and Forecasting model with Chemistry (WRF-Chem). The O3 pollution episode was generally well simulated by the WRF-Chem air quality model. On August 24, the high O3 levels with the

25 peak values of 250 μg m-3 occurred in the daytime mixing layer over the eastern YRD area. During nighttime, a shallow stable boundary layer formed near the surface, which decoupled the residual layer (RL) above it from the surface. O3 in the

decoupled RL remained nearly constant, resulting an O3-rich "reservoir", due to lack of NO titration and absence of dry deposition. The prevailing easterly wind in the lower troposphere transported the O3-rich air mass in the nocturnal RL from the eastern to western YRD. Consequently, the O3 concentrations in the RL over the western YRD area increased to 170 µg m-3 in the wee hours of August 25, 2016. Due to the growth of the convective boundary layer after the sunrise, entrainment of O3-rich RL air and boundary layer mixing contributed considerably to the rapid increase of surface O3. Process analysis indicated vertical mixing contributed ~40 µg m-3 h-1 of O3 accumulation over Nanjing in the morning of August 25, 2016, which played an important role in contributing to the severe daytime O3 pollution in the western YRD area. The mechanism of regional O3 transport through the nocturnal RL has a great implication for understanding O3 pollution in air quality change.

Key words: Tropospheric O<sub>3</sub>, residual layer, summer smog, regional O<sub>3</sub> transport, WRF-Chem

# 1 Introduction

Tropospheric O3 is an important atmospheric composition influencing climate change and air quality in different ways. According to the Intergovernmental Panel on Climate Change (IPCC, 2013), tropospheric O3 is one of the most important greenhouse gases. It is also a health hazard to sensitive individuals, reducing lung function and contributing to exacerbation of asthma symptoms (Bell et al., 2006). Tropospheric O3 is also important for atmospheric chemistry because its photolysis in the presence of water vapor is the primary source for hydroxyl radical (OH), which is responsible for the removal of many important trace gases. (Thompson, 1992; Logan et al., 1981).

The spatiotemporal variations of tropospheric O3 are substantial. In addition to photochemical reactions, atmospheric transports play an important role in determining the spatiotemporal distribution of tropospheric O3, including horizontal

transport (Wolff et al., 1977; Yienger et al., 2000; Wild and Akimoto, 2001; Lelieveld et al., 2002; Duncan et al., 2008; Liu et al., 2011; Zhu et al., 2017; Han et al., 2018) and vertical transport, e.g., exchange between stratosphere and troposphere (Hu et al., 2010; Jiang et al., 2015).

Ambient O3 levels are strongly influenced by diurnal variation of the atmospheric boundary layer (BL) structure. The daytime BL, also known as the convective boundary layer (CBL), is directly affected by solar heating of the earth's surface. In the

- major part of CBL, which is the mixing layer (ML), air pollutant concentrations distribute nearly uniformly resulted from the convective turbulent mixing. The nocturnal BL is often characterized by a stable layer (SL) near the surface and an overlying residual layer (RL). The SL develops due to radiative cooling after sunset. Above the SL, the remnants of the daytime ML form the RL with initially uniformly mixed air pollutants remaining from the preceding daytime (Stull, 1988). Nocturnal O3 in the RL could exert an impact on the ambient O3 variation during the following daytime (Aneja et al., 2000; Hu
- et al., 2012; Morris et al., 2010; Neu et al., 1994; Tong et al., 2011; Yorks et al., 2009; Hu et al., 2013; Klein et al., 2014). Locally, O3 from RL could contribute to the maximum surface O3 on the following day with estimating the enhancement of surface O3 by as much as 10-30 ppb (Hu et al., 2012). A few studies (Zhang et al., 1998; Zhang and Rao, 1999) investigated BL O3 episodes over the northeastern United States (U.S.) based on measurements and 1-D model, suggesting that O3 in the nocturnal RL could be transported to the downwind areas by the low-level jets over the eastern coast of U.S. Lee et al. (Lee et
- al., 2003) found that the daytime upslope flows transported O3 precursors up to the mountain, while the nocturnal downslope flow brought the O3-rich RL air mass downwards to Phoenix Valley, concluding that the transport, distribution and storage of O3 are highly impacted by background meteorological conditions. Zhang et al., (Zhang et al., 2015) found that a regional transport within the RL from the surrounding urban areas were contributing to a nighttime O3 peak on a mountaintop in East China. However, the regional O3 transport in the RL for air pollution has been incomprehensively understood especially for
- plain areas.

In recent years, ambient O3 levels have enhanced over the Yangtze River Delta (YRD) in East China with more frequent pollution events in spring and summer. Coupled with the increases of nitrogen oxide (NOx) and volatile organic compounds (VOCs) emissions, climate change of East Asian summer monsoon can significantly influence the O3 variations in the lower troposphere (An et al., 2015; Gao et al., 2016; Xu et al., 2008; Li et al., 2018). O3 pollution in the YRD could be attributed to

  - the most fitting meteorological condition for photochemistry and long-range transport of oxidants. Heat wave with less water vapor in the YRD contributed to less cloud cover and a strong solar radiation environment with significantly increasing photochemical reaction, potentially leading to substantial elevated O3 in a warmer climate (Tie et al., 2009; Li et al., 2012; Wang et al., 2017; Xie et al., 2016; Pu et al., 2017). It is therefore important to understand the formation mechanisms of O3 pollution including contributing meteorological factors of O3 pollution for summer smog over the YRD region.

This study focused on an O3 pollution episode observed over the YRD in August 2016. We aimed to explore the underlying mechanism on regional O3 transport over YRD by using observational data and WRF-Chem modeling. The rest of this paper was organized as follows: section 2 described the observational data and the pollution episode. Section 3 presented the WRF-Chem model methodology and validation. In section 4, a mechanism was explored to explain regional O3 transport in the RL from the eastern to western YRD. The conclusions were summarized in section 5.

#### **Observed O3 pollution episode** 80 2

# 2.1 Observation sites and data

Observation data of the Yangtze River Delta (YRD) urban sites of Nanjing (NJ), Zhenjiang (ZJ), Changzhou (CZ), Wuxi (WX), Suzhou (SZ) and Shanghai (SH) (Fig. 1) in August 2016 were used to study the pollution episode. The meteorological data were collected from China Meteorological Administration (CMA) and chemical data from the national environmental monitoring network of China. The meteorological data included wind speed (m  $s^{-1}$ ) and direction (deg.) at 10 m above ground level (AGL), air temperature ( $^{\circ}C$ ) and relative humidity (%) at 2 m AGL with a temporal resolution of 3 hrs, and total radiation irradiance with a time resolution of 1 hr. The chemical data were with a temporal resolution of 1 hr.

# 2.2 Summer smog in a heat wave episode over YRD

During a heat wave episode with the maximum temperature  $\geq$ 32 °C for 3 consecutive days over August 22-25, 2016, a summer smog with severe O<sub>3</sub> pollution occurred over the YRD region (Table 1) and high surface O<sub>3</sub> concentrations with the maximum 8 hr running mean values from 141.1 to 204.3 µg m<sup>-3</sup> were measured at the 6 urban sites of NJ, ZJ, CZ, WX, SZ and SH (Table 1). During this episode in mostly sunny days controlled by the westwards stretching subtropical anticyclone of the Western Pacific, the daily high air temperature kept from 32.8 to 34.0 °C over the YRD region with air temperature averaged during the O<sub>3</sub> maximum 8 hrs exceeding 32.0  $^{\circ}$ C. The O<sub>3</sub> concentrations over NJ of the western YRD were much 95 higher than the eastern YRD region (CZ, WX, SZ and SH) during this heat wave episode. It is generally accepted that high air temperature could cause strong photochemical reactions, producing high O<sub>3</sub> concentrations (Filleul et al., 2006; Pu et al., 2017; Seinfeld and Pandis, 1986).

## 2.3 A potential role of regional O3 transport

- Surface air temperature and solar radiation, deeply affect photochemical production. High levels of  $O_3$  concentrations were generally associated with high air temperatures (Rao et al., 1992; Council, 1991). However, we found from the observation of summer smog episode that the temperature and  $O_3$  levels exhibited the reversed changes from August 24 to 25 (Fig. 2 and Table 2). The maximum mean  $O_3$  concentrations increased from 230.1 µg m<sup>-3</sup> on August 24 to 284.8 µg m<sup>-3</sup> on August 25 2016 presenting an obvious enhancement in NJ of the western YRD, while the decreases in surface maximum total radiation irradiances and high air temperature respectively from 896 W m<sup>-2</sup> and 34.1 °C to 872 W m<sup>-2</sup> and 33.9 °C occurred during the
- two days. This was a noteworthy observational evidence with the increasing surface O<sub>3</sub> concentrations in the ambient air with decreasing daytime air temperature and total radiation irradiance from August 24 to 25 at NJ, an urban site of the western YRD (Figs. 1b-2, Table 2), which could be difficultly interpreted in the respect of photochemical production.
- It is believed that both strong local photochemical production and atmospheric transport can lead to high surface O<sub>3</sub> concentrations (Jacob, 1999; Carnero et al., 2010; Corsmeier et al., 1997; Gangoiti et al., 2002; Godowitch et al., 2011; Tang et al., 2017; Shu et al., 2016). Based on the available observation of gaseous species, it is estimated that the daily mean surface NO<sub>2</sub> concentrations varied slightly during August 24 and 25, reflecting a less impact of the local photochemical production on the daily enhanced O<sub>3</sub> from August 24 to 25. With excluding the impact of photochemical production, a potential role of regional O<sub>3</sub> transport over the YRD could become an important part in ambient O<sub>3</sub> pollution in the western YRD on August 25, 2016, which is explored with a modeling study in the following sections.

#### 115 **3** Simulation settings and validation

# **3.1** Simulation settings

To investigate the regional  $O_3$  transport over the YRD and the underlying mechanism, the Weather Research and Forecasting model with Chemistry (WRF-Chem) version 3.8.1 is employed (Grell et al., 2005) in this study. Three nested domains respectively with the horizontal resolutions of 45, 15 and 5 km cover the areas of East Asia, East China and YRD (Fig. 1) with 32 vertical layers extending from the surface to 100 hPa. The simulation period spans from 21 to 30 August 2016 with

1-hourly model outputs and the spin-up time of first 24 hours. The physical parameterizations include Noah land-surface model (Tewari et al., 2004), MM5 similarity surface layer, YSU boundary layer scheme (Hong et al., 2006), RRTM longwave scheme (Mlawer et al., 1997), Goddard shortwave scheme (Chou et al., 1998), Morrison double-moment microphysics scheme (Morrison et al., 2009), and Kain-Fritsch cumulus parameterization (Kain, 2004). The gas-phase chemical mechanism is the Regional Acid Deposition Model, version 2 (RADM2) (Chang et al., 1990; Stockwell et al., 1984) including 158 chemical reactions among 36 species. The NCEP Final Global Forecast System Operational Analysis (FNL) data is used to provide the initial and boundary conditions of meteorological variables for the WRF-Chem simulation. The chemical initial and lateral boundary conditions are extracted from the global chemical transport MOZART model (Model for Ozone And Related chemical Tracers) (Emmons et al., 2010; Horowitz et al., 2003). The Multi-resolution Emission Inventory for China (MEIC) (http://www.meicmodel.org/) of year 2012 is applied for the anthropocentric pollutant emissions, and the biogenic emissions are generated by the Model of Emissions of Gas and Aerosols from Nature (MEGAN) (Guenther

et al., 2006).

#### 3.2 Modeling validation

The simulations are compared with the wind speed, air temperature, relative humidity and O<sub>3</sub> concentrations observed at six
sites over the YRD (Fig. 1b) during August 22-25, 2016 for the O<sub>3</sub> pollution episode (Fig. 3). The correlation coefficients for near–surface air temperature and relative humidity reach up to 0.9 only with a slight overestimation of relative humidity. Over the sites NJ, CZ, WX, SZ and SH, the correlation coefficients between observed and simulated wind speed are above 0.6. The high correlation coefficients between the observed and simulated O<sub>3</sub> concentrations are ranged between 0.7 and 0.9 with small standard deviations, and their normalized root-mean-square (NRMS) are about 0.5. All the O<sub>3</sub> and meteorological
correlations have passed the significant level of 0.001 (except wind speed over ZJ, passing the significant level of 0.05). In general, the WRF-Chem simulated O<sub>3</sub>, air temperature, relative humidity and wind speed in the 6 cities of YRD show a good agreement with the observations. The simulation reasonably captures the observed changes of O<sub>3</sub> and meteorology during the summer smog episode over the YRD, which could be used to investigate the regional O<sub>3</sub> transport and the underlying mechanism over the YRD during the heat wave period in the following sections.

#### 145 4 Analysis on regional O3 transport

# 4.1 O3 "reservoir" in the RL

In order to analyze the development and evolution of  $O_3$  "reservoir" in the residual layer (RL) during the summer smog, the time-altitude cross sections of  $O_3$  concentrations and potential temperature over the western and eastern YRD region are chosen to present the temporal changes in the vertical structures of  $O_3$  concentrations and atmospheric boundary layer from August 24 to 25, 2016 based on the WRF-Chem modeling (Fig. 4).

Figure 4a presents the hourly changes of vertical  $O_3$  profiles from afternoon to midnight of August 24 over the eastern YRD region. In the afternoon of August 24, especially at 16:00 (local time, same for hereinafter), the surface  $O_3$  reached the peak concentrations of about 200 µg m<sup>-3</sup> in associated with the high air temperature during the heat wave, and the weak vertical gradients of potential temperature represented the well-developed mixing layer up to about 1.5 km height above the surface

for strong O<sub>3</sub> vertical mixing over the eastern YRD area (Fig. 4a). After the sunset, the near-surface O<sub>3</sub> was decreased sharply with ceased photochemical production and the O<sub>3</sub> consumption of dry deposition and NO titration, forming a typical O<sub>3</sub>-poor stable boundary layer and an overlying O<sub>3</sub> "reservoir" in the nocturnal RL over the eastern YRD region (Fig. 4a).

Figure 4b exhibits the hourly changes of vertical profiles of O<sub>3</sub> and potential temperature in the morning of August 25, 2016 over NJ of the western YRD region. Reflected with the strong vertical gradients of potential temperature, the existence of the stable boundary layer up to 0.1 km height over the surface in the nighttime prevented O<sub>3</sub>-rich air mass in the RL from vertical transport to the surface, building the O<sub>3</sub> "reservoir" in the RL from 0.1 km to 1 km height over the western YRD area (Fig 4b). After the sunrise on August 25, the stable boundary layer and RL vanished with the development of convective boundary layer (CBL), leading to the vertical mixing of O<sub>3</sub>-rich air mass in the RL and near-surface O<sub>3</sub>-poor air mass in the previous night for redistributing the O<sub>3</sub> concentrations in the daytime CBL (Fig. 4b).

When we compare the temporal changes of  $O_3$  "reservoir" in the nocturnal RL over the eastern and western YRD areas in Figures 4a and 4b, it is interesting that the eastern  $O_3$  "reservoir" obviously leaked with reducing the  $O_3$  concentrations in the RL over the nighttime of August 24 (Fig. 4a), while the western  $O_3$  "reservoir" was gradually strengthened with storing  $O_3$ -rich air mass in the nocturnal RL forming a high  $O_3$  center exceeding 180 µg m<sup>-3</sup> around 6 am before the sunrise on

August 25. Considering the prevailing easterly winds in the lower troposphere over the YRD region during the heat wave period, we could speculate a potential connection between the overnight decreases and increases of O<sub>3</sub> "reservoir" respectively over the eastern and western YRD areas (Figs. 4a and 4b) through regional O<sub>3</sub> transport in the nocturnal RL over the YRD, which are further investigated in the next sections to interpret the observational evidence of the exacerbated O<sub>3</sub> pollution in weakened photochemical production from August 24 to 25 in the western YRD (Figs. 1b-2, Table 2).

#### 4.2 O3 transport in the RL

- It is questionable why the nocturnal O<sub>3</sub> concentrations in the RL increased about 40 µg m<sup>-3</sup> over the western YRD region from 03:00 to 06:00 on August 25 (Fig. 4b). To investigate the regional O<sub>3</sub> transport over YRD with contributing to the O<sub>3</sub> enhancement in the nocturnal RL over the western YRD, Figure 5 presents the variations of O<sub>3</sub> concentrations and wind streamlines at the altitude of about 900 m in the RL in the morning on 25 over the YRD. It is clearly seen from Figure 5 that the prevailing easterly winds drove the O<sub>3</sub> transport from the eastern to western YRD region during the nighttime from August 24 to 25, confirming our speculation about the regional O<sub>3</sub> transport in the nocturnal RL over the YRD connecting
- between the overnight changes in  $O_3$  levels over the eastern and western areas (Figs. 4a and 4b). It is noteworthy that the regional  $O_3$  transport in the nocturnal RL reached westwards over the urban site NJ of the western YRD before the sunrise around 6:00 on August 25, in associated with the stagnation of cyclone circulation over NJ from 6:00 until 10:00, which prevented high  $O_3$  from moving further west and converged  $O_3$  into the RL above NJ and ZJ until the sunrise (Fig. 5).
- Figure 6 presents the temporal evolution of vertical sections of O<sub>3</sub> concentrations and atmospheric circulation along the regional O<sub>3</sub> transport over the YRD region to further explore the mechanism of regional O<sub>3</sub> transport over the YRD for summer smog. The vertical distributions of O<sub>3</sub> concentrations were controlled strongly by the diurnal change of BL structure. The daytime O<sub>3</sub> concentrations distributed vertically uniformly in the mixing layer (ML), the major part of CBL over the YRD (Fig. 6a), which could form the O<sub>3</sub>-rich RL after the sunset. During the nighttime, the O<sub>3</sub>-rich air mass in the RL were transported westwards from the eastern to western YRD, which were governed by the prevailing easterly winds in the lower troposphere (Figs. 6b-6e). Contributed by the nocturnal O<sub>3</sub> transport over the YRD, the RL with O<sub>3</sub>-rich air mass over the

western area was broken with the daytime ML development after the sunrise on August 25 (Fig. 6f). The daily large

contribution of vertical mixing to the surface  $O_3$  level occurred around 10:00 in the morning with downwards vertical mixing from the previous RL to the surface for summer smog (Figs. 6f and 7).

#### 195 4.3 Contribution of O3 vertical mixing from the RL

As discussed in sections 4.1 and 4.2, O<sub>3</sub>-rich air mass could transport from east to west in nocturnal RL over the YRD. As the convective boundary layer establishes after the sunrise during daytime, the O<sub>3</sub>-rich air mass could be entrained downwards to the surface (Mcelroy and Smith, 1993; Venkatram, 1977), contributing to the surface O<sub>3</sub> concentrations early in the day (Fig. 6f).

- Based on the WRF-Chem simulation, Figure 7 presents the hourly changes in the contribution rates of vertical mixing and local chemical reactions to surface  $O_3$  in the urban site NJ in the western YRD over August 24-25. Vertical mixing could be driven by vertical concentration gradients, and chemical reactions are net output of all O<sub>3</sub> chemical reactions (Gao et al., 2016). The positive and negative contribution rates indicate respectively the gain and loss of surface  $O_3$  concentrations through vertical mixing and local chemical reactions. The daily totals of positive contribution of vertical mixing and local 205 chemical reaction on August 24 and 25 are given in Table 3. Relatively to August 24, the positive contribution of O<sub>3</sub> vertical mixing enhanced significantly on August 25 with the largest contribution of about 40  $\mu$ g m<sup>-3</sup> h<sup>-1</sup>, twice as that on August 24 (Fig. 7). Tropospheric  $O_3$  results mainly from photochemical reactions in daytime (Seinfeld and Pandis, 1986). Although the largest contributions of chemical reactions reached up to 45  $\mu$ g m<sup>-3</sup> h<sup>-1</sup> and 53  $\mu$ g m<sup>-3</sup> h<sup>-1</sup> in the afternoon respectively on August 24 and 25 (Fig. 7), the daily totals of positive contributions of chemical reactions were estimated to be lower on August 25 with 298 µg m<sup>-3</sup> than the previous daytime of August 24 with 303 µg m<sup>-3</sup> in the western YRD area. The daily total 210 of positive O<sub>3</sub> contribution of vertical mixing raised sharply to 99  $\mu$ g m<sup>-3</sup> on August 25 with a large increase of 37  $\mu$ g m<sup>-3</sup> from August 24 (Table 3). The high O<sub>3</sub> levels in ambient air for summer smog in the western YRD on August 25 were significantly contributed from vertical mixing of O<sub>3</sub>-rich air mass transported in the previous nighttime RL from the eastern YRD. The regional O<sub>3</sub> transport in the nocturnal RL in associated with the diurnal changes of boundary layer are revealed to 215 be an important mechanism of regional O<sub>3</sub> transport in East China.

## 5 Conclusion

By analyzing the observational data of gaseous species and meteorological variables during severe summer smog over the YRD in East China in August, 2016, we found a noteworthy observational evidence with the increased daytime surface  $O_3$  concentrations in the ambient air of lower daytime air temperature and weaker solar radiation from August 24 to 25 in the western YRD with excluding the impact of photochemical production. Regional  $O_3$  transport over the YRD could play an important role in the ambient  $O_3$  pollution.

- By combining environmental and meteorological observation data with air quality modeling, the formation mechanism of  $O_3$  episode over the YRD area, particularly the extreme values over western YRD was investigated. On August 24, the high  $O_3$  levels peaked at about 250 µg m<sup>-3</sup> in the daytime mixing layer over the eastern YRD area. During nighttime, a shallow stable
- boundary layer formed near the surface, decoupled the RL above it with an O<sub>3</sub>-rich "reservoir". Governed by prevailing easterly wind in the lower troposphere, the O<sub>3</sub>-rich air mass in the nocturnal RL shifted from the eastern to western YRD. Consequently, the O<sub>3</sub> concentrations in the RL over the western YRD area enhanced up to 170 µg m<sup>-3</sup> in the early morning of August 25, 2016. In accompany with the growth of the convective boundary layer breaking up the RL after the sunrise, entrainment of O<sub>3</sub>-rich RL air and boundary layer mixing contributed considerably to the rapid increase of surface O<sub>3</sub>.
  Process analysis indicated vertical mixing contributed ~40 µg m<sup>-3</sup> h<sup>-1</sup> of O<sub>3</sub> accumulation over the western YRD in the

YRD.

This study discovered an important mechanism of regional  $O_3$  transport through the nocturnal RL from upstream to downstream areas driven by the prevailing winds in the lower troposphere in closely associated with the diurnal change of

morning of August 25, 2016, which was of great importance in formation of the severe daytime O<sub>3</sub> pollution in the western

235

220

atmospheric boundary layer, which could be depicted with a conceptual model in Figure 8. This mechanism of regional  $O_3$  transport has a substantial implication for understanding urban  $O_3$  pollution in air quality change.

The regional  $O_3$  transport in atmospheric boundary layer in this case of summer smog in the YRD, East China is to be further studied with more comprehensive observations of meteorology and environment as well as better modeling of the atmospheric boundary layer.