# Peer review of "An important mechanism of regional O3 transport for summer smog over the Yangtze River Delta in East China"

_Atmospheric Chemistry and Physics, 2018_

## Referee Comment (RC1) · Anonymous Referee #1 · 13 Aug 2018

General comments

In this work, the authors used WRF-Chem modelling system to simulate ozone and its precursors in YRD region in China, and analyzed the mechanism of regional ozone transport in a severe photochemical pollution episode. The combination of observation data and mode simulation illustrates the important mechanism of O3 transport from the upstream to the downstream through the residual layer in the Yangtze River Delta region, which is of great significance for understanding the summer daytime O3 pollution. The manuscript is well organized and the methodology is feasible, and it may be of great interest to the China's ozone modelers and the local governments. How-

ever, there are some problems in the simulation and discussion (as shown below). I recommend the publication of journal ACP after the problems were clarified.

Specific comments

1. Abstract should state briefly the purpose of the research, the principal results and major conclusions. Therefore, I suggest the author to rewrite the abstract, avoiding some unimportant statements.

2. The validation of the vertical profile of ozone (or column ozone concentrations) is very important in the analysis of ozone budget, but missing in this study. The reviewer suggests that the evaluation of ozone characteristics and budget should be conducted using not only surface measurements but also aircraft and/or column measurements.

3. In addition to temperature and radiation, wind direction and speed also important meteorological factors for ozone pollution. I recommend the authors adding the statistics of wind in Table 2 and interpreting the difference in the manuscript.

4. Vertical mixing and chemical production are two main factors for ozone difference between 24 and 25 Aug. However, the authors forgot the other factor – dry deposition. I recommend the authors showing the difference of dry deposition in different days.

5. As the authors hypothesize horizontal transport in the residual layer is the main reason for the ozone pollution in 25 Aug. I recommend the authors analyzing the horizontal contribution in the RL carefully using the process analysis.

Technical comments

1. I recommend changing summer smog to photochemical smog throughout the manuscript.

2. Please check all the subscript and superscript throughout the manuscript.

3. Please check all the abbreviation throughout the manuscript. All the abbreviation should be interpreted in abstract and main article separately. Generally, if the phrase

used more than three times, it can be defined by abbreviation. Otherwise, please use the full name of the phrase. For example, AGL has not been used more than three times.

---

## Referee Comment (RC2) · Anonymous Referee #2 · 13 Aug 2018

General comments

In this study, using the observations of surface ozone concentrations and some meteorological parameters and WRF-Chem model simulations, the authors analyzed the mechanism of regional ozone transport in a severe summer smog episode in the YRD region of China. This work revealed the fact that ozone can be transported from the upstream to the downstream through residual layer during nighttime and then from the residual layer to the surface due to the convective turbulence in the ABL after the sunrise, which is an important mechanism of ozone transport and the formation of ozone pollution in the near surface layer. The manuscript is well organized and easily under-

stood. I recommend the publication of journal ACP after minor revision.

Specific comments

1. Please explain the definition of high air temperature in Table 2. Also, in Line 93, what is the daily high air temperature? Do you mean the daily maximum air temperature?

2. The authors mentioned the effect of dry deposition and NOx titration to O3 consumption several times in the study when analyzing the change of surface O3 concentration. Could you show the temporal variation of dry deposition and NOx concentration during the O3 pollution episode in NJ?

3. Please add the statistics of wind speed and direction in Table 2 and explain the difference in the manuscript.

Technical comments

1. Please check all the subscript and superscript throughout the manuscript.

---

## Referee Comment (RC3) · Anonymous Referee #3 · 23 Aug 2018

General comments

This manuscript, using chemistry transport model WRF-Chem results to investigate one of typical summer ozone episodes observed in the Yangtze River Delta Region (YRD) in Eastern China. The model results was validated by meteorological and air quality observational data. The specific ozone episode was characterized by the nocturnal ozone transport over the residual layer (RL) and the daytime vertical mixing process. The decoupled RL holds the ozone produced from daytime and redistribute ozone concentration there due to large-scale circulations. The ozone-rich air mass from RL can be touch surface and enhance surface ozone levels by strong daytime

[Figure]

CBL mixing processes. The authors clarify the ozone episode with the important mechanism. The results are very interesting and the study is meaningful for understanding the formation of the high ozone episodes in the YRD region. I recommend its publication in a revision in accordance with the following review comments.

Major comments

1. Maximum 8-hour ozone is used in observational ozone analysis, for example in Table 1. However, the model results are hourly-basis. That may lead to some mismatches in the description of the variation of ozone concentrations because the diurnal cycle of hour-ozone and 8-hour ozone are different and time of peak values is shifted from each other.

2. Diurnal cycle of Temperature, solar radiation, hourly, and 8-hour ozone are peaked in different time. It is hard to directly relate those parameters with 8-hour ozone concentrations.

3. Different high air temperature and maximum total radiation described in Table 2 may lead to significant difference of biogenic VOC emissions. I agree with you about the anthropogenic emission can be considered as constant during the episode. But the impact of changes in BVOC on variation of ozone concentrations may also need to be checked.

Minor comments

Line 59 on Page 3: " . . . by the downwind the low-level jets over the eastern coast of U.S. Lee et al" might be " . . . by the downwind the low-level jets over the western coast of U.S. Lee et al"

Line 79 on Page 4: "WRF-Chem model methodology and validation . . ." is better to change into "WRF-Chem modelling methodology and model validation . . ."

Line 91 on Page 4: "maximum 8-hr running mean values" should be "maximum 8-hr running mean values".

Line 68 on Page 3: I do not understand how the large-scale and long-term climate change of East Asian summer monsoon can significantly influence the surface ozone variations like this two days episode.

Line 84 and 87 on Page 4: change "the chemical data" into "the air quality monitoring data".

Line 142-144 on Page 5: Re-write "...The simulation reasonable ....in the following section" because it is hard to be understood.

Line 170-173 on Page 8: please re-write this paragraph for more clear.

―――――――――――――――――

---

## Referee Comment (RC4) · Anonymous Referee #4 · 24 Aug 2018

Comment:

The authors presented their efforts in applying observations and model simulation to analyze a severe O3 pollution case in China. This is an important and interesting topic considering the adverse health effect of O3. The materials are reasonably organized, and the unique horizontal transport and vertical mixing mechanisms were reported. Therefore I would recommend this manuscript to be published if the following concerns can be properly addressed.

Major comment:

(1). Please consider rephrase the whole manuscript for English editing with help from

native speaker. There are a lot grammar errors, confusing lengthy sentences, and improper wordings. Some are pointed out in the minor comments. The current shape is not acceptable for scientific journal publication.

(2). The unique transport and vertical mixing mechanism shall be discussed with more in-depth analysis, including: first, background introduction is necessary to briefly describe the general condition of O3 and meteorology over the study domain, thus the findings from examining the extreme event can be highlighted. For example, multi-year data of local O3 observations (or satellite products) could be used to demonstrate the frequency, seasonality, and spatial distribution of O3 smog in YRD. Climate data (e.g., observation from NCDC or China Meteorology Agency) could also be used to describe the general PBL condition in YRD; Second, discussion about the transport shall be improved with more solid demonstrations, many of the current statements were roughly made without sufficient proofs. For example, section2 promotes the hypothesis that the extreme O3 event was due to regional transport, yet no discussion was made to exclude the potential impact from local emission or photochemical production; Third, the most important one, the driving forces of the unique transport and mixing processes were not discussed at all. The authors spent a lot efforts to describe the severe O3 case and how it was accumulated though regional transport, but paid little attention to the causes. For example, in Fig.4(a), why O3 in 0-0.5km was depleted after 20:00, but remained high in 0.5-1km? The near surface layer NO might be responsible for titration but no demonstrate was made; Does the residual layer present in all seasons, and does it always host O3 or other atmospheric pollutants? The southeast wind shown in this study seems closely related with East Asia summer monsoon, thus does it also carry excessive O3 from the ocean into inland YRD? Fundamental questions such as what make the high O3 concentration in residual layer remained unsolved. These are the key findings that shall be reported in a journal publication.

Minor comment:

(1). P2-L27: Spell "NO" before use it.

(2). P3-L64: change word "incomprehensively"

(3). P3-L68: This manuscript has no in-depth discussion of the "climate change of Asian summer monsoon" or its impact on O3, I would suggest remove this sentence or add the related discussion

(4). A brief introduction of the typical O3 concentration urban areas of China would be necessary, to clarify if the high O3 in YRD is an area-dependent condition or a national wide issue.

(5). Table 1 & Figure 1: Are there multiple sites or is there only one site for each city? Please also provide the web source or reference for the observation data

(6). P4-L85: Why wind speed is collected at 10m but temperature and relative humidity are collected at 2m? For evaluation purpose, WRF can output wind speed at both 10m & 2m, and NCDC has observation data for both too.

(7). P4-L95-97: Do you try to compare Temperature & O3 between western (NJ) and eastern YRD? Local emissions would be another factor determining O3, the conclusion made in line#95-96 was made without solid demonstration.

(8). P4-L95: "The O3 concentrations over NJ of the western YRD were much higher ..." this is not professional scientific writing, please describe it with exact numbers.

(9). P5-L99: "Surface air temperature and solar radiation, deeply affect photochemical production." Please rewrite this sentence or remove it, these are unnecessary common sense for journal publication.

(10). P5-L101: "exhibited" shall be "showed" ?

(11). P5-L102-105: Please rewrite this lengthy sentence, either break it into a few short ones or rephrase.

(12). P5-L103: "NJ of the western YRD" this term has been used several times in the manuscript, I would recommend simply using "NJ" or "the western part of YRD".

(13). Fig.2 & Table2: Why the data from other sites were not shown?

(14). P5-L108: Unnecessary, in addition to local production and transport, what else can result in high O3?

(15). P5-L110: "it is estimated that the daily mean surface NO2 concentrations varied slightly during August 24 and 25 ". Analysis of NO2 is important and necessary to be included as it supports your conclusion.

(16). P6-L130: Latest MEIC updates the emission to 2015, if the 2012 emission was not projected to 2016, it's better to rerun the simulation with latest emission inputs.

(17). P6-L134: Incorrect grammar, it shall be "Simulated wind speed, air temperature, relative humidity, and O3 concentrations are compared with observations . . . "

(18). Section3.2: More evaluation statistics, such as normalized mean bias and root mean square error shall be applied to demonstrate model performance. Fig.3 cannot tell the absolute values of simulation bias. P6-L120-125 listed details of model configuration but no reason was given to clarify why these options were selected. It's also necessary to briefly compare the simulation performance with other published WRF-Chem applications over YRD region.

(19). P7-L145: "Analysis on" shall be "Analyzing" or "Analysis of"

(20). P7-L153: It's necessary to include a brief introduction of the climatology in NJ area before using "heat wave".

(21). Fig.4: Need a clear definition of "eastern" and "western" if you are showing subdomain averages in the figure.

(22). P7-L165: Please rewrite this lengthy and confusing sentence.

(23). P8-L175: Please change the word "questionable", check it in the dictionary before using it.

[Figure]

(24). Fig.5: No prominent changes of O3 or wind stream are shown, why use 4 sub-panels?

(25). Fig.6 cross sections are drawn along the red line in Fig.1. If the observation along this track is not discussed, I would recommend to make cross-sectional figures along the travel path in Fig.5.

(26). P9-L201: Please specify how "vertical mixing" is calculated, if it is directly output by WRF-Chem, a bar chart would be better for Fig.7 to present the contributions from all processes.

---

## Author Response (AR1)

Dear editors and four reviewers:

Thank you all for your comments concerning our manuscript entitled "An important mechanism of regional $O_3$ transport for summer smog over the Yangtze River Delta in East China" (Manuscript ID: acp-2018-479). Those comments are all valuable and very helpful for revising and improving manuscript. We have studied comments carefully and have accordingly made the revisions. The revised parts are highlighted with Track Changes in the revised manuscript. In the following we quoted each review question in the square brackets and added our response after each paragraph.
* * *
**For Referee #1:**

Many thanks for your encouraging comments. We have revised the manuscript accordingly. Furthermore, following the suggestion of reviewer #4, we have rerun the simulation with the latest **MEIC** emission inventories of 2015 and analyzed the updated simulation over YRD in the revised manuscript, although there are small differences of $O_3$ simulation over the YRD region between MEIC emissions 2012 and 2015. All the revisions have been highlighted with Track Changes in the revised manuscript. The point-by-point responses to the reviewer's comments are as follows:

*General comments:*
*1.* *"In this work, the authors used WRF-Chem modelling system to simulate ozone and its precursors in YRD region in China, and analyzed the mechanism of regional ozone transport in a severe photochemical pollution episode. The combination of observation data and mode simulation illustrates the important mechanism of O3 transport from the upstream to the downstream through the residual layer in the Yangtze River Delta region, which is of great significance for understanding the summer daytime O3 pollution. The manuscript is well organized and the methodology is feasible, and it may be of great interest to the China's ozone modelers and the local governments. However, there are some problems in the simulation and discussion (as shown below). I recommend the publication of journal ACP after the problems were clarified."*

**Response 1:** Thanks for the reviewer's positive comments on our manuscript. We have revised carefully the manuscript following the reviewer's comments.

*Specific comments:*
*1.* *"Abstract should state briefly the purpose of the research, the principal results and major conclusions. Therefore, I suggest the author to rewrite the abstract, avoiding some unimportant statements."*

**Response 1:** Following the reviewer's comments, we have rewritten the abstract in the revised mmanuscript.

*2.* *"The validation of the vertical profile of ozone (or column ozone concentrations) is very important in the analysis of ozone budget, but missing in this study. The reviewer suggests that the evaluation of ozone characteristics and budget should be conducted using not only surface measurements but also aircraft and/or column measurements."*

**Response 2:** We agree with the reviewer's suggestion. The validation of the vertical profile of $O_3$ (or column $O_3$ concentrations) is very important in the analysis of $O_3$ budget. However, the observation data of $O_3$ vertical profile (or column $O_3$ concentrations) over YRD during this pollution episode are not available for us to evaluate the vertical structure of $O_3$ from simulation. In the revised manuscript, we have added the following discussions in the last paragraph of 3.2 Modeling Validation (section 3.2):

The validation of the vertical structures of $O_3$ is very important in the analysis of $O_3$ budget, but unavailable for us to evaluate the vertical structure of $O_3$ from simulation. If there would be observational data of $O_3$ vertical profiles, the validation of vertical profiles of $O_3$ could be done in future study of $O_3$ budget.

**3.** *"In addition to temperature and radiation, wind direction and speed also important meteorological factors for ozone pollution. I recommend the authors adding the statistics of wind in Table 2 and interpreting the difference in the manuscript."*

**Response 3:** Following the reviewer's suggestions, we have added wind speed and direction in Table 2 and the corresponding discussions (section 2.3 (paragraph 2)) in the revised manuscript as follows:

The near-surface easterly winds prevailed in the directions of 90 deg. and 111 deg. with the daily averaged wind speeds of 2.4 and 2.6 m s$^{-1}$ respectively on August 24 and 25 at NJ (Table 2), indicating the fewer changes in both wind speed and direction over NJ during those two days.

**Table 2: Meteorological and environmental elements observed at an urban site NJ of the western YRD from August 24 to 25, 2016 with their daily differences ($\Delta$x).**

|  | Aug. 24 | Aug. 25 | $\Delta$x |
|---|---|---|---|
| Maximum 8-hour running mean surface $O_3$ concentrations ($\mu$g m$^{-3}$) | 230.1 | 284.8 | 54.7 |
| Maximum hourly surface $O_3$ concentration ($\mu$g m$^{-3}$) | 256.8 | 317.2 | 60.4 |
| Daytime mean surface $O_3$ concentrations ($\mu$g m$^{-3}$) | 180.6 | 230.1 | 49.5 |
| Daytime mean surface $NO_2$ concentrations ($\mu$g m$^{-3}$) | 27.9 | 27.8 | - 0.1 |
| Daily maximum air temperature at 2 m (℃) | 34.1 | 33.9 | - 0.2 |
| Maximum surface total radiation irradiance (W m$^{-2}$) | 896.0 | 872.0 | - 24.0 |
| Daytime mean surface total radiation irradiance (W m$^{-2}$) | 511.8 | 423.4 | - 88.4 |
| Daily mean wind speed at 10 m (m s$^{-1}$) | 2.4 | 2.6 | 0.2 |
| Daily mean wind direction at 10 m (deg.) | 90 | 111 | 21 |

**4.** *"Vertical mixing and chemical production are two main factors for ozone difference between 24 and 25 Aug. However, the authors forgot the other factor – dry deposition. I recommend the authors showing the difference of dry deposition in different days."*

**Response 4:** Thanks for reviewer's comments.

Based on the modeling, we have calculated the hourly changes of $O_3$ dry depositions (Fig. S1) and estimated the daily averages of dry deposition rates with about 0.42 and 0.49 $\mu g \ m^{-2} \ s^{-1}$ respectively for August 24 and 25. The dry depositions of $O_3$ varied little over these two days with a slight enhancement on August 25, reflecting $O_3$ dry depositions exerted less impact on surface $O_3$ change during August 24-25. The contribution of $O_3$ dry deposition to tropospheric $O_3$ changes was trivial compared to vertical mixing and chemical reactions (Wang et al., 1998; Fowler et al., 1999; Zavier et al., 2003).

We have added the above discussions in the revised manuscript (section 4.3 (paragraph 2)).

[Figure]

**Figure S1. Hourly changes of $O_3$ dry deposition flux in NJ, an urban area of the western YRD during August 24 (08-24) and 25 (08-25).**

References:

Wang, Y., Logan, J. A., and Jacob, D. J.: Global simulation of tropospheric $O_3$-NOx-hydrocarbon chemistry: 2. Model evaluation and global ozone budget, Journal of Geophysical Research Atmospheres, 103, 10713-10725, 1998.

Fowler, D., Cape, J., Coyle, M., Smith, R., Hjellbrekke, A.-G., Simpson, D., Derwent, R., and Johnson, C.: Modelling photochemical oxidant formation, transport, deposition and exposure of terrestrial ecosystems, Environmental Pollution, 100, 43-55, 1999.

Zaveri, R. A., Berkowitz, C. M., Kleinman, L. I., Springston, S. R., Doskey, P. V., Lonneman, W. A., and Spicer, C. W.: Ozone production efficiency and NOx depletion in an urban plume: Interpretation of field observations and implications for

**5.** *"As the authors hypothesize horizontal transport in the residual layer is the main reason for the ozone pollution in 25 Aug. I recommend the authors analyzing the horizontal contribution in the RL carefully using the process analysis."*

**Response 5:** Thanks for reviewer's comments. We have accordingly calculated the $O_3$ transport flux of NJ from the eastern YRD region based on the process analysis. We have added the analysis in the revised manuscript (paragraph 2 of section 4.2):

The $O_3$ transport flux in the nocturnal RL over the YRD region was calculated based on the process analysis. It was estimated that the $O_3$ horizontal transport flux in RL averaged over the nighttime from 20:00 on August 24 to 8:00 on 25 was 541 µg m$^{-2}$ s$^{-1}$ at the western site NJ with 119 µg m$^{-2}$ s$^{-1}$ stronger than that during the preceding night to August 24, reflecting the larger contribution of $O_3$ horizontal transport in RL to the $O_3$ pollution on August 25 over the western YRD.

*Technical comments:*

**1.** *"I recommend changing summer smog to photochemical smog throughout the manuscript."*

**Response 1:** This study is mostly focused on the analysis on physical process of regional $O_3$ transport. To avoid the confusion with the photochemical process, we have kept "summer smog" with some changes to "photochemical smog" in the revised manuscript.

**2.** *"Please check all the subscript and superscript throughout the manuscript."*

**Response 2:** We have checked and corrected all the subscripts and superscripts throughout the manuscript.

**3.** *"Please check all the abbreviation throughout the manuscript. All the abbreviation should be interpreted in abstract and main article separately. Generally, if the phrase used more than three times, it can be defined by abbreviation. Otherwise, please use the full name of the phrase. For example, AGL has not been used more than three times."*

**Response 3:** Thanks for the careful edition of reviewer. We have checked and corrected these errors throughout the manuscript.

**For Referee #2:**

Many thanks for your encouraging comments. We have revised the manuscript accordingly. Furthermore, following the suggestion of reviewer #4, we have rerun the simulation with the latest **MEIC** emission inventories of 2015 and analyzed the updated simulation over YRD in the revised manuscript, although there are small differences of $O_3$ simulation over the YRD region between MEIC emissions 2012 and 2015. All the revisions have been highlighted with Track Changes in the revised manuscript. The point-by-point responses to the reviewer's comments are as follows:

**General comments:**

*1.* *"In this study, using the observations of surface ozone concentrations and some meteorological parameters and WRF-Chem model simulations, the authors analyzed the mechanism of regional ozone transport in a severe summer smog episode in the YRD region of China. This work revealed the fact that ozone can be transported from the upstream to the downstream through residual layer during nighttime and then from the residual layer to the surface due to the convective turbulence in the ABL after the sunrise, which is an important mechanism of ozone transport and the formation of ozone pollution in the near surface layer. The manuscript is well organized and easily understood."*

**Response 1:** We appreciate the reviewer's positive comments on our manuscript. We have revised carefully the manuscript based on the following comments.

**Specific comments:**

*1.* *"Please explain the definition of high air temperature in Table 2. Also, in Line 93, what is the daily high air temperature? Do you mean the daily maximum air temperature?"*

**Response 1:** Thanks for reviewer's comments. The values of high air temperature (in Table 2) are the daily maximum air temperature at 2 m. We have clarified that in the revised manuscript.

*2.* *"The authors mentioned the effect of dry deposition and NOx titration to $O_3$ consumption several times in the study when analyzing the change of surface $O_3$ concentration. Could you show the temporal variation of dry deposition and NOx concentration during the $O_3$ pollution episode in NJ?"*

**Response 2:** Thanks for reviewer's comments. The contribution rates of chemical reactions to surface $O_3$ are minus over nighttime (Fig. 7) due to NOx titration. During the nighttime from August 24 to 25, the average consumption of chemical reactions was about -8.0 $\mu g$ $m^{-3}$ $h^{-1}$, while it was -8.5 $\mu g$ $m^{-3}$ $h^{-1}$ over the preceding nighttime to August 24.

Following the reviewer's comments, we have presented the hourly changes of dry deposition fluxes of $O_3$ during the $O_3$ pollution episode in NJ and added the following discussions in the revised manuscript (section 4.3 (paragraph 2)):

Based on the modeling, we have calculated the hourly changes of $O_3$ dry depositions (Fig. S1) and estimated the daily averages of dry deposition rates with about 0.42 and 0.49 µg m$^{-2}$ s$^{-1}$ respectively for August 24 and 25. The dry depositions of $O_3$ varied little over these two days with a slight enhancement on August 25, reflecting $O_3$ dry depositions exerted less impact on surface $O_3$ change during August 24-25. The contribution of $O_3$ dry deposition to tropospheric $O_3$ changes was trivial compared to vertical mixing and chemical reactions (Wang et al., 1998; Fowler et al., 1999; Zavier et al., 2003).

[Figure]

**Figure S1. Hourly changes of $O_3$ dry deposition flux in NJ, an urban area of the western YRD during August 24 (08-24) and 25 (08-25).**

**Reference:**
Wang, Y., Logan, J. A., and Jacob, D. J.: Global simulation of tropospheric $O_3$-NOx-hydrocarbon chemistry: 2. Model evaluation and global ozone budget, Journal of Geophysical Research Atmospheres, 103, 10713-10725, 1998.

Fowler, D., Cape, J., Coyle, M., Smith, R., Hjellbrekke, A.-G., Simpson, D., Derwent, R., and Johnson, C.: Modelling photochemical oxidant formation, transport, deposition and exposure of terrestrial ecosystems, Environmental Pollution, 100, 43-55, 1999.

Zaveri, R. A., Berkowitz, C. M., Kleinman, L. I., Springston, S. R., Doskey, P. V., Lonneman, W. A., and Spicer, C. W.: Ozone production efficiency and NOx depletion in an urban plume: Interpretation of field observations and implications for evaluating O3-NOx-VOC sensitivity, Journal of Geophysical Research: Atmospheres, 108, 2003.

**3.** *"Please add the statistics of wind speed and direction in Table 2 and explain the difference in the manuscript."*

**Response 3:** Following the reviewer's suggestions, we have added wind speed and direction in Table 2 and the discussions (paragraph 2 of section 2.3) in the revised manuscript as follows:

The near-surface easterly winds prevailed in the directions of 90 deg. and 111 deg. with the daily averaged wind speeds of 2.4 and 2.6 m s$^{-1}$ respectively on August 24 and 25 at NJ (Table 2), indicating the fewer changes in both wind speed and direction over NJ during those two days.

**Table 2: Meteorological and environmental elements observed at an urban site NJ of the western YRD from August 24 to 25, 2016 with their daily differences ($\Delta$x).**

|  | Aug. 24 | Aug. 25 | $\Delta$x |
|---|---|---|---|
| Maximum 8-hour running mean surface $O_3$ concentrations ($\mu$g m$^{-3}$) | 230.1 | 284.8 | 54.7 |
| Maximum hourly surface $O_3$ concentration ($\mu$g m$^{-3}$) | 256.8 | 317.2 | 60.4 |
| Daytime mean surface $O_3$ concentrations ($\mu$g m$^{-3}$) | 180.6 | 230.1 | 49.5 |
| Daytime mean surface $NO_2$ concentrations ($\mu$g m$^{-3}$) | 27.9 | 27.8 | - 0.1 |
| Daily maximum air temperature at 2 m ($^\circ$C) | 34.1 | 33.9 | - 0.2 |
| Maximum surface total radiation irradiance (W m$^{-2}$) | 896.0 | 872.0 | - 24.0 |
| Daytime mean surface total radiation irradiance (W m$^{-2}$) | 511.8 | 423.4 | - 88.4 |
| Daily mean wind speed at 10 m (m s$^{-1}$) | 2.4 | 2.6 | 0.2 |
| Daily mean wind direction at 10 m (deg.) | 90 | 111 | 21 |

**Technical comments:**

**1.** *"Please check all the subscript and superscript throughout the manuscript."*

**Response 1:** We have checked and corrected all the subscripts and superscripts throughout the manuscript.

**For Referee #3:**

Many thanks for your encouraging comments. We have revised the manuscript accordingly. Furthermore, following the suggestion of reviewer #4, we have rerun the simulation with the latest **MEIC** emission inventories of 2015 and analyzed the updated simulation over YRD in the revised manuscript, although there are small differences of $O_3$ simulation over the YRD region between MEIC emissions 2012 and 2015. All the revisions have been highlighted with Track Changes in the revised manuscript. The point-by-point responses to the reviewer's comments are as follows:

**General comments:**

*1.    "This manuscript, using chemistry transport model WRF-Chem results to investigate one of typical summer ozone episodes observed in the Yangtze River Delta Region (YRD) in Eastern China. The model results was validated by meteorological and air quality observational data. The specific ozone episode was characterized by the nocturnal ozone transport over the residual layer (RL) and the daytime vertical mixing process. The decoupled RL holds the ozone produced from daytime and redistribute ozone concentration there due to large-scale circulations. The ozone-rich air mass from RL can be touch surface and enhance surface ozone levels by strong daytime CBL mixing processes. The authors clarify the ozone episode with the important mechanism. The results are very interesting and the study is meaningful for understanding the formation of the high ozone episodes in the YRD region. I recommend its publication in a revision in accordance with the following review comments."*

**Response 1:** We appreciate the reviewer's positive comments on our manuscript. And we have revised carefully the manuscript based on the following comments.

**Specific comments:**

*1.    "Maximum 8-hour ozone is used in observational ozone analysis, for example in Table 1. However, the model results are hourly-basis. That may lead to some mismatches in the description of the variation of ozone concentrations because the diurnal cycle of hour-ozone and 8-hour ozone are different and time of peak values is shifted from each other."*

**Response 1:** Thanks for reviewer's comments. In the ambient air quality standards, the standard of $O_3$ pollution (photochemical smog or summer smog) is defined with the maximum 8 hour running mean of $O_3$ concentrations. According to the standard of $O_3$ pollution, we analyzed the maximum 8 hour running mean of $O_3$ concentrations to only identify the $O_3$ pollution episode over YRD (Table 1) in sections 2.2 and 2.3. Hourly $O_3$ concentration was used to analyze the diurnal cycle over NJ (Fig. 2). To better validate the modelling, we compared the hourly changes of observed and simulated $O_3$ concentrations. Based on the hourly data of $O_3$ simulation, we discussed the diurnal cycle of hourly $O_3$ concentration and $O_3$ transport. We have added the description to avoid the misleading in the revised manuscript (sections 2.2 and 2.3).

**2.** *"Diurnal cycle of Temperature, solar radiation, hourly, and 8-hour ozone are peaked in different time. It is hard to directly relate those parameters with 8-hour ozone concentrations."*

**Response 2:** Thanks for reviewer's comments. We have used the maximum 8 hour running mean $O_3$ concentrations to only identify the $O_3$ pollution episode over YRD, and used hourly $O_3$ concentration to represent $O_3$ diurnal cycle related with solar radiation and temperature. We have added the description to avoid the misleading in the revised manuscript (sections 2.2 and 2.3).

**3.** *"Different high air temperature and maximum total radiation described in Table 2 may lead to significant difference of biogenic VOC emissions. I agree with you about the anthropogenic emission can be considered as constant during the episode. But the impact of changes in BVOC on variation of ozone concentrations may also need to be checked."*

**Response 3:** We agree with reviewer's suggestion. Different high air temperature and maximum total radiation may lead to a significant difference of biogenic VOC (BVOC) emissions. The impact of changes in BVOC on $O_3$ concentrations would be done in future study with available data of BVOC emissions.

The above discussion has been added in the conclusions of revised manuscript (last paragraph).

**Minor comments:**

**1.** *"Line 59 on Page 3: '... by the downwind the low-level jets over the eastern coast of U.S. Lee et al' might be ' ... by the downwind the low-level jets over the western coast of U.S. Lee et al'."*

**Response 1:** Thanks for this suggestion. We have checked that it was over the eastern coast of U.S.

**2.** *"Line 79 on Page 4: 'WRF-Chem model methodology and validation ...' is better to change into 'WRF-Chem modelling methodology and model validation ...'."*

**Response 2:** It has been changed in the revised manuscript.

**3.** *"Line 91 on Page 4: 'maximum 8-hr running mean values' should be 'maximum 8-hr running mean values'."*

**Response 3:** It has been corrected as follows (section 2.2):

During a heat wave episode with the maximum temperature ≥32 °C for 3 consecutive days over August 22-25, 2016, a summer smog with severe $O_3$ pollution occurred over the YRD region (Table 1) and high surface $O_3$ concentrations with the averages of maximum 8-hour running mean values from 141.1 to 204.3 μg m$^{-3}$ were measured at the 6 urban sites of NJ, ZJ, CZ, WX, SZ and SH (Table 1), which exceed the second national primary standard of ambient air quality standards (100 μg m$^{-3}$).

**4.** *"Line 68 on Page 3: I do not understand how the large-scale and long-term climate change of East Asian summer monsoon can significantly influence the surface ozone variations like this two days episode."*

**Response 4:** Thanks for comments. We have deleted the "the large-scale and long-term" there in the revised manuscript.

**5.** *"Line 84 and 87 on Page 4: change 'the chemical data' into 'the air quality monitoring data'."*

**Response 5:** It has been changed in the revised manuscript.

**6.** *"Line 142-144 on Page 5: Re-write '...The simulation reasonable ....in the following section' because it is hard to be understood."*

**Response 6:** Thanks for comments. We have rewritten these sentences as follows:

The simulation reasonably captures the observed changes of $O_3$ and meteorology during the summer smog episode over the YRD. Therefore, the simulation data could be used to investigate the regional $O_3$ transport and the underlying mechanism over the YRD during the summer smog period, as presented in the following sections.

**7.** *"Line 170-173 on Page 8: please re-write this paragraph for more clear."*

**Response 7:** Following the comments. We have rewritten those sentences as follows ( the last paragraph of section 4.1 ):

Considering the prevailing easterly winds in the lower troposphere over the YRD region during the summer smog period, we could speculate that the regional $O_3$ transport in the nocturnal RL could connect between the eastern decreases and NJ increases of overnight $O_3$ "reservoir" over the YRD region (Figs. 4a and 4b). We further investigated that the regional $O_3$ transport in the nocturnal RL over the YRD to interpret the observational evidence of the exacerbated $O_3$ pollution in weaker photochemical production on August 25 in the NJ site of western YRD (Figs. 1b-2, Table 2).

**For Referee #4:**
Many thanks for your encouraging comments. We have revised the manuscript accordingly. All the revisions have been highlighted with Track Changes in the revised manuscript. The point-by-point responses to the reviewer's comments are as follows:

**General comments:**

*1.* *"The authors presented their efforts in applying observations and model simulation to analyze a severe O3 pollution case in China. This is an important and interesting topic considering the adverse health effect of O3. The materials are reasonably organized, and the unique horizontal transport and vertical mixing mechanisms were reported. Therefore I would recommend this manuscript to be published if the following concerns can be properly addressed."*

**Response:** We appreciate the reviewer's positive comments on our manuscript. We have revised carefully the manuscript based on the following comments.

**Major comment:**

*1.* *"Please consider rephrase the whole manuscript for English editing with help from native speaker. There are a lot grammar errors, confusing lengthy sentences, and improper wordings. Some are pointed out in the minor comments. The current shape is not acceptable for scientific journal publication."*

**Response:** Thanks for reviewer's comments. We have rephrased the whole manuscript for English editing with the help of native speaker in modifying grammar errors, confusing lengthy sentences and improper wordings. Please find all the revisions highlighted with Track Changes in the revised manuscript.

*2.* *"The unique transport and vertical mixing mechanism shall be discussed with more in-depth analysis, including:"*
*1)* *"first, background introduction is necessary to briefly describe the general condition of O3 and meteorology over the study domain, thus the findings from examining the extreme event can be highlighted. For example, multi-year data of local O3 observations (or satellite products) could be used to demonstrate the frequency, seasonality, and spatial distribution of O3 smog in YRD. Climate data (e.g., observation from NCDC or China Meteorology Agency) could also be used to describe the general PBL condition in YRD; "*

**Response 1):** Thanks for your suggestion. The according introduction has been added in the revised manuscript as follows (section 1 (paragraph 5)):

In recent years, ambient $O_3$ levels have enhanced over the Yangtze River Delta (YRD) in East China with more frequent pollution events from late May to July (Tang et al., 2013). During 1990 to 2013, the hourly $O_3$ peaks varied from 140 to 167 ppbv (about 294-350 $\mu g\ m^{-3}$) in the YRD region, from 160 to 180 ppbv (about 336-378 $\mu g\ m^{-3}$) in the Beijing-Tianjin-Hebei area over North China Plain and

from 200 to 220 ppbv (about 420-462 µg m$^{-3}$) in the Pearl River Delta (Wang et al., 2017). Coupled with the increases of nitrogen oxides (NO$_x$) and volatile organic compounds (VOCs) emissions, O$_3$ distribution in the lower troposphere is significantly influenced by winds, air temperature, cloud cover, and downward shortwave radiation through changing the transport and chemical formation of O$_3$ (An et al., 2015; Gao et al., 2016; Xu et al., 2008; Li et al., 2018). O$_3$ levels could increase with a rate of 4–5 ppb K$^{-1}$ when temperature was between 28 and 38 °C (Pu et al., 2017). The prevailing winds driving transport of air pollutants from the YRD industrialized areas might have contributed to the O$_3$ enhancement (Tang et al., 2013). The ambient O$_3$ levels could be affected by the diurnal variation of atmospheric BL structure over YRD with nighttime stable BL height at 200 m and the daytime BL height reaching up to about 1200 m (Chang et al., 2016).

References:

Tang, H., Liu, G., Zhu, J., Han, Y., and Kobayashi, K.: Seasonal variations in surface ozone as influenced by Asian summer monsoon and biomass burning in agricultural fields of the northern Yangtze River Delta, Atmospheric research, 122, 67-76, 2013.

Wang, T., Xue, L., Brimblecombe, P., Lam, Y. F., Li, L., and Zhang, L.: Ozone pollution in China: A review of concentrations, meteorological influences, chemical precursors, and effects, Science of the Total Environment, 575, 1582-1596, 2017.

An, J., Zou, J., Wang, J., Lin, X., and Zhu, B.: Differences in ozone photochemical characteristics between the megacity Nanjing and its suburban surroundings, Yangtze River Delta, China, Environmental Science and Pollution Research, 22, 19607, 2015.

Gao, J., Zhu, B., Xiao, H., Kang, H., Hou, X., and Shao, P.: A case study of surface ozone source apportionment during a high concentration episode, under frequent shifting wind conditions over the Yangtze River Delta, China, Science of the Total Environment, 544, 853, 2016.

Xu, X., Lin, W., Wang, T., Yan, P., Tang, J., Meng, Z., and Wang, Y.: Long-term trend of surface ozone at a regional background station in eastern China 1991–2006: enhanced variability, Atmospheric Chemistry and Physics, 8, 2595-2607, 2008.

Li, S., Wang, T., Huang, X., Pu, X., Li, M., Chen, P., Yang, X. Q., and Wang, M.: Impact of East Asian summer monsoon on surface ozone pattern in China, Journal of Geophysical Research: Atmospheres, 123, 1401-1411, 2018.

Pu, X., Wang, T., Huang, X., Melas, D., Zanis, P., Papanastasiou, D., and Poupkou, A.: Enhanced surface ozone during the heat wave of 2013 in Yangtze River Delta region, China, Science of the Total Environment, 603, 807-816, 2017.

Chang, Y., Zou, Z., Deng, C., Huang, K., Collett, J. L., Lin, J., and Zhuang, G.: The importance of vehicle emissions as a source of atmospheric ammonia in the megacity of Shanghai, Atmospheric Chemistry and Physics, 16, 3577, 2016.

**2)** *"Second, discussion about the transport shall be improved with more solid demonstrations, many of the current statements were roughly made without sufficient proofs. For example, section2 promotes the hypothesis that the extreme O3 event was due to regional transport, yet no discussion was made to exclude the potential impact from local emission or photochemical production; "*

**Response 2):** Thanks for reviewer's comments. Following the comments, we have improved discussion about the transport in the revised manuscript as follows (sections 2.3 and 4.3):

Tropospheric $O_3$ levels are controlled by regional transports of $O_3$ and the precursors as well as photochemical production in closely associated with local emissions of $O_3$ precursors. Considering weak changes of local emissions in short time, the WRF-Chem simulation with the hourly emissions of chemical species over YRD unchanged between August 24 and 25. To analyze the impact from photochemical production, we used the surface $NO_2$ concentrations and total radiation irradiance (TRI) to analyze the change of photochemical production rates. There were no apparent changes of $NO_2$ and TRI between August 24 and 25, indicating that photochemical production exerted less impact on the high $O_3$ level on August 25 compared to regional $O_3$ transport in nocturnal RL. The analysis of simulation results, revealed that vertical mixing from the upper $O_3$-rich RL to daytime surface layer was a large contributor to $O_3$ enhancement on August 25 (Fig. 7).

**3)** *"Third, the most important one, the driving forces of the unique transport and mixing processes were not discussed at all. The authors spent a lot efforts to describe the severe O3 case and how it was accumulated though regional transport, but paid little attention to the causes. For example, in Fig.4(a), why O3 in 0-0.5km was depleted after 20:00, but remained high in 0.5-1km? The near surface layer NO might be responsible for titration but no demonstrate was made; Does the residual layer present in all seasons, and does it always host O3 or other atmospheric pollutants? The southeast wind shown in this study seems closely related with East Asia summer monsoon, thus does it also carry excessive O3 from the ocean into inland YRD? Fundamental questions such as what make the high O3 concentration in residual layer remained unsolved. These are the key findings that shall be reported in a journal publication."*

**Response :** Thanks for comments. Here are the responses for each of comments.

**a)** *"Third, the most important one, the driving forces of the unique transport and mixing processes were not discussed at all. The authors spent a lot efforts to describe the severe O3 case and how it was accumulated though regional transport, but paid little attention to the causes."*

**Response a):** Thanks for comments. According to your suggestions, we have revised the manuscript as follows (sections 4.2 and 4.3):

Under the guidance of the prevailing easterly winds, the $O_3$ transport from the eastern to western YRD region persisted during the nighttime from August 24 to 25, confirming our speculation about the regional $O_3$ transport in the nocturnal RL over the YRD region (Figs. 4a and 4b). As the regional $O_3$ transport reached the RL over the western YRD, $O_3$ concentrations in the RL accumulated up to 200 μg m$^{-3}$ over the western site NJ around 6 am in the sunrise hours of August 25. In accompany with the disappearance of the residual layer after sunrise, the vertical mixing initiated by convective and turbulent processes in the development of daytime convective boundary layer. The vertical mixing in the convective boundary layer after sunrise from the upper levels to the ground with the net downward transport flux reaching up to 35 μg m$^{-3}$ h$^{-1}$, contributing a considerable surface $O_3$ accumulation to the $O_3$ pollution during summer smog on August 25 in the western YRD region (Fig. 7).

**b)** *"For example, in Fig. 4(a), why O3 in 0-0.5km was depleted after 20:00, but remained high in 0.5-1km? The near surface layer NO might be responsible for titration but no demonstrate was made;"*

**Response b):** Based on the current understanding of atmospheric chemistry in boundary layer (BL), ambient $O_3$ levels are strongly influenced by diurnal variation of the (BL) structure. The daytime BL, also known as the convective boundary layer (CBL), is directly affected by solar heating of the earth's surface. In the major part of CBL, which is the mixing layer (ML), air pollutant concentrations distribute nearly uniformly resulted from the convective turbulent mixing. The nocturnal BL is often characterized by a stable layer (SL) near the surface and an overlying residual layer (RL). The SL develops due to radiative cooling after sunset. Above the SL, the remnants of the daytime ML form the RL with initially uniformly mixed air pollutants remaining from the preceding daytime (Stull, 1988), and $O_3$ is a representative remnant in the RL with the lack of $O_3$ consumption of NO titration and dry deposition during nighttime (Xie et al., 2016; Sillman, 1999). Over the Eastern YRD, the SL at 0-0.5km and the RL at 0.5-1.0km were developed after 20:00 on August 24, leading to the $O_3$ depletion in 0-0.5km and but remained high in 0.5-1 km overnight.

In section 4.3 (Fig. 7), we calculated the contribution rates of overnight chemical reactions to surface $O_3$, which could represent the nocturnal consumption of NO titration with the negative contribution rates. During the nighttime from August 24 to 25, the average consumption of chemical reactions was about -8.0 μg m$^{-3}$ h$^{-1}$, while it was -8.5 μg m$^{-3}$ h$^{-1}$ over the preceding nighttime to August 24.

References:

Stull, R. B.: An introduction to boundary layer meteorology, Atmospheric Sciences Library, 8, 89, 1988.

Xie, M., Zhu, K., Wang, T., Chen, P., Han, Y., Li, S., Zhuang, B., and Shu, L.: Temporal characterization and regional contribution to $O_3$ and NOx at an urban and a suburban site in Nanjing, China, Science of the Total Environment, 551, 533-545, 2016.

Sillman, S., 1999. The relation between ozone, NOx and hydrocarbons in urban and polluted rural environments. Atmos. Environ. 33, 1821–1845.

**c)** *"Does the residual layer present in all seasons, and does it always host O3 or other atmospheric pollutants? Fundamental questions such as what make the high O3 concentration in residual layer remained unsolved."*

**Response c):** Yes, the residual layer with could generally present in all seasons (Morris et al., 2010; Venzac et al., 2009; Neu et al., 1994) excepting the strong disturbation of atmospheric circulation in free troposphere. The residual layer could host $O_3$ or other atmospheric pollutants depending on changes of atmospheric pollutants and the interaction with atmospheric boundary layer.

Based on fundamental theory of atmospheric chemistry in boundary layer (BL), the daytime BL, also known as the convective boundary layer (CBL), is directly affected by solar heating of the earth's surface. In the major part of CBL, which is the mixing layer (ML), air pollutant concentrations distribute nearly uniformly resulted from the convective turbulent mixing. The nocturnal BL is often characterized by a stable layer (SL) near the surface and an overlying residual layer (RL). The SL develops due to radiative cooling after sunset. Above the SL, the remnants of the daytime ML form the RL with initially uniformly mixed air pollutants remaining from the preceding daytime (Stull, 1988), and $O_3$ is a representative remnant in the RL with the lack of $O_3$ consumption of NO titration and dry deposition during nighttime (Xie et al., 2016; Sillman, 1999).

References:

Morris, G. A., Ford, B., Rappenglück, B., Thompson, A. M., Mefferd, A., Ngan, F., and Lefer, B.: An evaluation of the interaction of morning residual layer and afternoon mixed layer ozone in Houston using ozonesonde data, Atmospheric Environment, 44, 4024-4034, 2010.

Venzac, H., Sellegri, K., Villani, P., Picard, D., and Laj, P.: Seasonal variation of aerosol size distributions in the free troposphere and residual layer at the puy de Dôme station, France, Atmospheric Chemistry and Physics, 9, 1465-1478, 2009.

Neu, U., Künzle, T., and Wanner, H.: On the relation between ozone storage in the residual layer and daily variation in near-surface ozone concentration—a case study, Boundary-Layer Meteorology, 69, 221-247, 1994.

Stull, R. B.: An introduction to boundary layer meteorology, Atmospheric Sciences Library, 8, 89, 1988.

Xie, M., Zhu, K., Wang, T., Chen, P., Han, Y., Li, S., Zhuang, B., and Shu, L.: Temporal characterization and regional contribution to $O_3$ and NOx at an urban and a suburban site in Nanjing, China, Science of the Total Environment, 551, 533-545, 2016.

Sillman, S., 1999. The relation between ozone, NOx and hydrocarbons in urban and polluted rural environments. Atmos. Environ. 33, 1821–1845.

**d)** *"The southeast wind shown in this study seems closely related with East Asia summer monsoon, thus does it also carry excessive O3 from the ocean into inland YRD?"*

**Response d):** We agree with the reviewer's comments. The southeast winds, which are closely related with East Asian summer monsoon, could carry excessive $O_3$ from the ocean into inland YRD. The influencing extension and strength of $O_3$ import from ocean to land could be a further study on air quality in YRD, especially for the coastal areas.

**Minor comment:**

**1.** *"P2-L27: Spell "NO" before use it."*
**Response 1 :** It has been done.

**2.** *"P3-L64: change word "incomprehensively"*
**Response 2:** We have changed it to "poorly".

**3.** *"P3-L68: This manuscript has no in-depth discussion of the "climate change of Asian summer monsoon" or its impact on O3, I would suggest remove this sentence or add the related discussion"*
**Response 3:** Thanks for suggestion. We have removed this sentence.

**4.** *"A brief introduction of the typical O3 concentration urban areas of China would be necessary, to clarify if the high O3 in YRD is an area-dependent condition or a national wide issue."*

**Response 4:** Following the reviewer's suggestion, we have added the discussion in manuscript as follows (Lines 71-77):

In recent years, ambient $O_3$ levels have enhanced over the Yangtze River Delta (YRD) in East China with more frequent pollution events from late May to July (Tang et al., 2013). During 1990 to 2013, the hourly $O_3$ peaks varied from 140 to 167 ppbv (about 294-350 µg m$^{-3}$) in the YRD region, from

160 to 180 ppbv (about 336-378 μg m$^{-3}$) in the Beijing-Tianjin-Hebei area over North China Plain and from 200 to 220 ppbv (about 420-462 μg m$^{-3}$) in the Pearl River Delta (Wang et al., 2017).

References:

Tang, H., Liu, G., Zhu, J., Han, Y., and Kobayashi, K.: Seasonal variations in surface ozone as influenced by Asian summer monsoon and biomass burning in agricultural fields of the northern Yangtze River Delta, Atmospheric research, 122, 67-76, 2013.

Wang, T., Xue, L., Brimblecombe, P., Lam, Y. F., Li, L., and Zhang, L.: Ozone pollution in China: A review of concentrations, meteorological influences, chemical precursors, and effects, Science of the Total Environment, 575, 1582-1596, 2017.

**5.** *"Table 1 & Figure 1: Are there multiple sites or is there only one site for each city? Please also provide the web source or reference for the observation data"*

**Response 5:** The surface O$_3$ concentrations were averaged from multiple sites for each city, and meteorological variables were only one site for each city. We have provided the web sources for the observation data in the revised manuscript as follows (section 2.1):

The meteorological data were collected from China Meteorological Administration (http://www.cnemc.cn) and the air quality monitoring data from the national environmental monitoring network of China (http://www.mep.gov.cn). The meteorological data were observed at meteorological sites of each city (one site for each city), including wind speed (m s$^{-1}$) and direction (deg.) at 10 m above ground level, air temperature (°C) and relative humidity (%) at 2 m above ground level with a temporal resolution of 3 hours, and total radiation irradiance with a time resolution of 1 hour. The air quality monitoring data used in the paper were the mean value of multiple sites in each city, with the temporal resolution of 1 hour.

**6.** *"P4-L85: Why wind speed is collected at 10m but temperature and relative humidity are collected at 2m? For evaluation purpose, WRF can output wind speed at both 10m & 2m, and NCDC has observation data for both too."*

**Response 6:** According to the World Meteorological Organization (WMO) standards, wind speed at 10 m and air temperature and relative humidity at 2 m are conventionally observed in the global weather monitoring network. Therefore, we have output wind speed and direction at 10 m, air temperature and relative humidity at 2 m to compare with observations.

Thanks for the information from reviewer. The NCDC observation data of meteorology could be used for further study.

**7.** *"P4-L95-97: Do you try to compare Temperature & O3 between western (NJ) and eastern YRD? Local emissions would be another factor determining O3, the conclusion made in line#95-96 was made without solid demonstration."*

**Response 7:** We have compared statistics of air temperature and O$_3$ changes at 6 YRD sites based on the hourly observation data in Table 1. Local emissions would be another factor determining the

ambient $O_3$ levels. However, there are usually less changes in local emissions from day to day. Therefore, we could exclude the impact of local emission change on the high $O_3$ level on August 25.

The conclusion made in line#95-96 has been revised as follows: It is generally accepted that high $O_3$ concentrations are accompanied by high air temperature with strong photochemical reactions (Filleul et al., 2006; Pu et al., 2017; Seinfeld and Pandis, 1986). Pu et al (2017) found that $O_3$ level increases with a rate of 4-5 ppb $K^{-1}$ when temperature is between 28 and 38 ℃ in NJ of YRD (section 1 (paragraph 4)).

**Table 1: Averages (Ave) of maximum 8-hour running mean surface $O_3$ concentrations (μg $m^{-3}$), mean air temperature (℃) over the periods of maximum 8-hour running mean surface $O_3$ concentrations and daily maximum air temperature (℃) their standard deviations (Std) over August 22-25, 2016 at 6 YRD sites.**

| Sites | Maximum 8-hour running mean $O_3$ | | Mean air temperature | | Daily maximum air temperature | |
|-------|-----|-----|-----|-----|-----|-----|
|       | Ave | Std | Ave | Std | Ave | Std |
| NJ | 204.3 | 58.2 | 32.6 | 0.5 | 33.4 | 0.7 |
| ZJ | 163.3 | 44.7 | 32.6 | 0.7 | 33.2 | 0.8 |
| CZ | 174.4 | 34.8 | 33.2 | 0.7 | 34.2 | 0.9 |
| WX | 190.5 | 24.9 | 33.4 | 0.3 | 34.5 | 0.6 |
| SZ | 173.7 | 21.0 | 33.3 | 0.2 | 34.0 | 0.2 |
| SH | 141.1 | 7.8 | 32.0 | 0.4 | 32.7 | 0.3 |

References:

Filleul, L., Cassadou, S., Médina, S., Fabres, P., Lefranc, A., Eilstein, D., Tertre, A. L., Pascal, L., Chardon, B., and Blanchard, M.: The Relation between Temperature, Ozone, and Mortality in Nine French Cities during the Heat Wave of 2003, Environmental Health Perspectives, 114, 1344, 2006.

Pu, X., Wang, T., Huang, X., Melas, D., Zanis, P., Papanastasiou, D., and Poupkou, A.: Enhanced surface ozone during the heat wave of 2013 in Yangtze River Delta region, China, Science of the Total Environment, 603, 807-816, 2017.

Seinfeld, J. H., and Pandis, S. N.: Atmospheric chemistry and physics: From air pollution to climate change (Second Edition), Wiley, 1595-1595 pp., 1986.

**8.** *"P4-L95: 'The O3 concentrations over NJ of the western YRD were much higher…' this is not professional scientific writing, please describe it with exact numbers."*

**Response 8:** Thanks for suggestion. We have revised as follows (section 2.2):

The $O_3$ concentrations over NJ of the western YRD were 10-63 μg $m^{-3}$ higher than the eastern YRD region (CZ, WX, SZ and SH) during this heat wave episode.

**9.** *"P5-L99: 'Surface air temperature and solar radiation, deeply affect photochemical production.' Please rewrite this sentence or remove it, these are unnecessary common sense for journal publication."*

**Response 9:** It has been removed.

**10.** *"P5-L101: 'exhibited' shall be 'showed'?"*
**Response 10:** It has been revised.

**11.** *"P5-L102-105: Please rewrite this lengthy sentence, either break it into a few short ones or rephrase."*

**Response 11:** It has been rewritten as follows (section 2.3 (paragraph 1)):

The maximum 8-hour running mean $O_3$ concentrations increased from 230.1 µg m$^{-3}$ on August 24 to 284.8 µg m$^{-3}$ on August 25 2016, and maximum hourly $O_3$ concentrations enhanced from 256.8 µg m$^{-3}$ on August 24 to 317.2 µg m$^{-3}$ on August 25 2016, presenting an obvious enhancement in western urban site NJ. In contrast, surface maximum total radiation irradiances and maximum air temperature respectively decreased from 896 W m$^{-2}$ and 34.1 ℃ to 872 W m$^{-2}$ and 33.9 ℃ during the two days.

**12.** *"P5-L103: 'NJ of the western YRD' this term has been used several times in the manuscript, I would recommend simply using 'NJ' or 'the western part of YRD'."*
**Response 12:** We have changed it to "NJ" there.

**13.** *"Fig.2 & Table2: Why the data from other sites were not shown?"*

**Response 13:** We have showed the statistics on the observational data of all the YRD sites in Table 1. Following review's suggestion, here we present the hourly changes of $O_3$, air temperature, wind speed and direction in six cities in the following Figures S1, S2 and S3.

[Figure]

**Fig. S1 Hourly series of surface O₃ concentrations in NJ, ZJ, CZ, WX, SZ and SH.**

[Figure]

**Fig. S2 Time series of 2 m air temperature in NJ, ZJ, CZ, WX, SZ and SH.**

[Figure]

**Fig. S3 Time series of 10 m wind speed and direction in NJ, ZJ, CZ, WX, SZ and SH.**

**14.** *"P5-L108: Unnecessary, in addition to local production and transport, what else can result in high O3?"*

**Response 14:** It has been modified.

**15.** *"P5-L110: 'it is estimated that the daily mean surface NO2 concentrations varied slightly during August 24 and 25'. Analysis of NO2 is important and necessary to be included as it supports your conclusion."*

**Response 15:** Thanks for your comments. We agree with the comments.   Analysis of NO2 is important and necessary to be included as it supports our conclusion.

**16.** *"P6-L130: Latest MEIC updates the emission to 2015, if the 2012 emission was not projected to 2016, it's better to rerun the simulation with latest emission inputs."*

**Response 16:** Following the reviewer's suggestion,we have rerun the simulation with the latest *MEIC* emission inventories of 2015 and analyzed the updated simulation over YRD in the revised manuscript, although there are small differences of $O_3$ simulation over the YRD region between MEIC emissions 2012 and 2015.

**17.** *"P6-L134: Incorrect grammar, it shall be 'Simulated wind speed, air temperature, relative humidity, and O3 concentrations are compared with observations…'"*

**Response 17:** We have revised as your suggestion as follows (section 3.2):

Simulated wind speed, air temperature, relative humidity, and $O_3$ concentrations are compared with observations at six sites over the YRD (Fig. 1b) during August 22-25, 2016 for the $O_3$ pollution episode (Fig. 3).

**18.** *"Section3.2: More evaluation statistics, such as normalized mean bias and root mean square error shall be applied to demonstrate model performance. Fig.3 cannot tell the absolute values of simulation bias. P6-L120-125 listed details of model configuration but no reason was given to clarify why these options were selected. It's also necessary to briefly compare the simulation performance with other published WRF-Chem applications over YRD region."*

**Response 18:** Thanks reviewer's suggestion. We have calculated these statistics in Table. S1. The simulation reasonably captures the observed changes of $O_3$ and meteorology during the summer smog episode over the YRD.

We have briefly compared with the previous WRF-Chem studies over YRD region (Table S2), to optimize the simulation configurations in our study, and the simulation result in manuscript had a good performance compared those studies' result

Table S1 MB, NMB and RMSE of $O_3$ ($\mu g$ $m^{-3}$), wind speed (m $s^{-1}$), temperature (℃) and relative humidity (%) between simulation and observations.

|     |      | $O_3$ | Wind Speed | Temperature | Relative Humidity |
|-----|------|-------|------------|-------------|-------------------|
|     | MB   | -18.7 | 0.7 | -0.1 | -1.3 |
| NJ  | NMB  | -17.6% | 25.5% | -4.3% | -2.1% |
|     | RMSE | 36.8 | 1.3 | 1.5 | 9.7 |
|     | MB   | -12.6 | 2.1 | -0.8 | 0.5 |
| ZJ  | NMB  | -12.8% | 108.7% | -2.8% | 0. 7% |
|     | RMSE | 29.5 | 2.6 | 1.7 | 8.7 |
|     | MB   | -12.5 | 1.2 | -2.2 | 6. 6 |
| CZ  | NMB  | -13.4% | 47.9% | -8.2% | 10.0% |
|     | RMSE | 30.1 | 1.6 | 2.9 | 11.2 |
|     | MB   | -22.9 | 1.1 | -2.4 | 11.1 |
| WX  | NMB  | -22.1% | 47.0% | -8.6% | 18.0% |
|     | RMSE | 37.2 | 1.6 | 2.8 | 13.7 |
|     | MB   | -18.1 | 1.3 | -2.6 | 12.4 |
| SZ  | NMB  | -17.9% | 59.0% | -9.2% | 19.5% |
|     | RMSE | 34.6 | 1.7 | 3.0 | 14.6 |
|     | MB   | -23.4 | 2.0 | -1.2 | 8.2 |
| SH  | NMB  | -21.9% | 68.0% | -4.3% | 12.7% |
|     | RMSE | 31.9 | 2.6 | 1.7 | 9.7 |

MB ( mean bias) $= [\sum_{i=1}^{n}(S_i - O_i)]/n$

NMB (normalized mean bias) $= [\sum_{i=1}^{n}(S_i - O_i)]/(\sum_{i=1}^{n} O_i) * 100\%$

RMSE (root mean square error) $= \sqrt{[\sum_{i=1}^{n}(S_i - O_i)^2]/n}$

Where Si and Oi are the simulated and observed value

Table S2 Model configuration in some other references.

| Item | Our work | Gao et al | Zhang | Wang | Liao | Xie | Zhong |
|------|----------|-----------|-------|------|------|-----|-------|
| Microphysics scheme | Morrison | Lin et al | Lin et al | NCEP-5 | Lin et al | Lin et al | Morrison |
| Long wave radiation | RRTM | RRTM | RRTM | RRTM | RRTM | RRTM | RRTMG |
| Short wave radiation | Goddard | Goddard | Dudhia | Dudhia | Goddard | Goddard | RRTMG |
| Boundary layer | YSU | YSU | YSU | YSU | YSU | MYJ | MYJ |
| Land surface | Noah | Noah | Noah | Noah | Noah | Noah | Noah |
| Cumulus physics | Kain-Fritsch | Grell 3D | Kain-Fritsch | Kain-Fritsch | Kain-Fritsch | Kain-Fritsch | None |
| Gas-phase chemical mechanism | RADM2 | CBM-Z | CBM-Z | RADM2 | CBM-Z | CBM-Z | RADM2 |

**References**

Gao, J., Zhu, B., Xiao, H., Kang, H., Hou, X., and Shao, P.: A case study of surface ozone source apportionment during a high concentration episode, under frequent shifting wind conditions over the Yangtze River Delta, China, Science of the Total Environment, 544, 853, 2016.

Zhang, L., Jin, L., Zhao, T., Yan, Y., Zhu, B., Shan, Y., Guo, X., Tan, C., Gao, J., and Wang, H.: Diurnal variation of surface ozone in mountainous areas: Case study of Mt. Huang, East China, Science of the Total Environment, 538, 583-590, 2015.

Wang, X., Chen, F., Wu, Z., Zhang, M., Tewari, M., Guenther, A., and Wiedinmyer, C.: Impacts of weather conditions modified by urban expansion on surface ozone: comparison between the Pearl River Delta and Yangtze River Delta regions, Advances in Atmospheric Sciences, 26, 962-972, 2009.

Liao, J., Wang, T., Jiang, Z., Zhuang, B., Xie, M., Yin, C., Wang, X., Zhu, J., Fu, Y., and Zhang, Y.: WRF/Chem modeling of the impacts of urban expansion on regional climate and air pollutants in Yangtze River Delta, China, Atmospheric Environment, 106, 204-214, 2015.

Xie, M., Liao, J., Wang, T., Zhu, K., Zhuang, B., Han, Y., Li, M., and Li, S.: Modeling of the anthropogenic heat flux and its effect on air quality over the Yangtze River Delta region, China, Atmospheric Chemistry & Physics Discussions, 15, 2015.

Zhong, S., Qian, Y., Zhao, C., Leung, R., Wang, H., Yang, B., Fan, J., Yan, H., Yang, X.-Q., and Liu, D.: Urbanization-induced urban heat island and aerosol effects on climate extremes in the Yangtze River Delta region of China, Atmospheric Chemistry and Physics, 17, 5439-5457, 2017.

**19.** *"P7-L145: 'Analysis on' shall be 'Analyzing' or 'Analysis of'"*
**Response 19:** We have changed it to "Analysis of".

**20.** *"P7-L153: It's necessary to include a brief introduction of the climatology in NJ area before using "heat wave"."*
**Response 20:** We have revised and added introduction of climatology in YRD and NJ in manuscript as follows (section 2.2):

With the Western Pacific subtropical high staying over the YRD region, heat wave events with high surface air temperature occur over this region in summer. The averages of daily temperature and daily maximum temperature in YRD were 27.1 and 39.5 ℃ in summer during 2013-2016, and the average of summer daily temperature was 28.5 ℃ in NJ.

**21.** *"Fig.4: Need a clear definition of 'eastern' and 'western' if you are showing subdomain averages in the figure."*
**Response 21:** In the manuscript, the western YRD covered the site NJ, and the eastern YRD included CZ, WX, SZ and SH.

**22.** *"P7-L165: Please rewrite this lengthy and confusing sentence."*
**Response 22:** We have revised these (section 4.1 (paragraph 4)):

We compared the temporal changes of $O_3$ "reservoir" in the nocturnal RL over the wstern and eastern YRD areas in Figures 4a and 4b. It is interesting that the eastern $O_3$ "reservoir" obviously leaked with reducing the $O_3$ concentrations over the nighttime of August 24 (Fig. 4a), while the western $O_3$ "reservoir" was gradually strengthened, forming a high $O_3$ center exceeding 200 µg m$^{-3}$ around 6 am on August 25.

**23.** *"P8-L175: Please change the word "questionable", check it in the dictionary before using it."*
**Response 23:** We have changed it to "worth discussing".

**24.** *"Fig.5: No prominent changes of O3 or wind stream are shown, why use 4 subpanels?"*

**Response 24:** Fig. 5 shows the growth of cyclone circulation over NJ from 00:00 to 09:00. The center of cyclone circulation moved from southwest to south of NJ, and the wind direction changed from east-southeast to southeast at 900 m in residual layer. Hence, due to the cyclone circulation over NJ from 06:00 until 09:00, high $O_3$ converged in the RL over NJ and ZJ at the sunrise (Fig. 5).

**25.** *"Fig.6 cross sections are drawn along the red line in Fig.1. If the observation along this track is not discussed, I would recommend to make cross-sectional figures along the travel path in Fig.5."*

**Response 25:** Fig. 5 is focused at cyclone circulation over NJ and ZJ in western YRD from 00:00 to 09:00 on August 25. Fig. 1 shows the complete YRD region containing both western and eastern YRD region. Since the cross section in Fig.6 shows the full scope from east to west during August 24-25, it is better to use the red line in Fig.1 to show the travel path.

**26.** *"P9-L201: Please specify how "vertical mixing" is calculated, if it is directly output by WRF-Chem, a bar chart would be better for Fig.7 to present the contributions from all processes."*

**Response 26:** Thanks for comments. The contribution from "vertical mixing" is output by WRF-Chem. Actually, we have tried to plot a bar chart for Fig. 7, and the line-symbol figure was better to express temporal changes.

**An important mechanism of regional $O_3$ transport for summer smog over the Yangtze River Delta in East China**

Jun Hu[1], Yichen Li[2], Tianliang Zhao[1,*], Jane Liu[2,3], Xiao-Ming Hu[4], Duanyang Liu[5,] Yongcheng Jiang[1,6], Jianming Xu[7], Luyu Chang[7]

[1] Collaborative Innovation Center on Forecast and Evaluation of Meteorological Disasters, Key Laboratory for Aerosol-Cloud-Precipitation of China Meteorological Administration, Nanjing University of Information Science & Technology,Nanjing 210044, China

[2] School of Atmospheric Sciences, Nanjing University,Nanjing,210046, China

[3] University of Toronto, Toronto, M5S 3G3, Canada

[4] Center for Analysis and Prediction of Storms, and School of Meteorology, University of Oklahoma, Norman, Oklahoma 73072, USA

[5] Jiangsu Meteorological Observatory, Nanjing, 210008, China

[6] Laboratory of Strait Meteorology, Xiamen Meteorological Observatory, Xiamen Meteorological Bureau, Xiamen 361012, China

[7] Yangtze River Delta Center for Environmental Meteorology Prediction and Warning, Shanghai 200030, China

*Correspondence to*: Tianliang Zhao (tlzhao@nuist.edu.cn)

**Abstract**. Severe ozone ($O_3$) pollution episodes plague a few regions in Eastern China at times, e.g., the Yangtze River Delta (YRD). The formation mechanisms including  meteorological factors contributing these severe pollution events remain elusive. A severe summer smog stretched over the YRD region from August 22 to 25, 2016 with hourly surface $O_3$ concentrations exceeding 300 μg m$^{-3}$ on August 25 in Nanjing,  an urban area in the western YRD. The weather pattern

of this episode was characterized by near-surface prevailing easterly wind and continuous high air temperature. The formation mechanism of this $O_3$ episode over the YRD area, particularly the extreme values over the western YRD, was investigated by using observation data and simulation with the Weather Research and Forecasting model with Chemistry (WRF-Chem).  The results showed that the extremely

25    high surface $O_3$ concentration  in the western YRD area  on August 25 was  largely contributed by regional $O_3$ transport in the nocturnal residual layer (RL) with the diurnal change of atmospheric boundary layer.  On August 24, the high $O_3$ levels with the peak values of  220 µg m$^{-3}$ occurred in the daytime mixing layer over the eastern YRD area. During nighttime from August

30    24 to 25, a shallow stable boundary layer formed near the surface, which decoupled the RL above it from the surface. $O_3$ in the decoupled RL remained nearly constant, resulting in an $O_3$-rich "reservoir" in the RL with  the lack of $O_3$ consumption from nitrogen oxide (NO) titration and dry deposition during nighttime . The prevailing easterly wind in the lower troposphere governed the regional transport of $O_3$-rich air mass in the nocturnal RL from the eastern to western YRD. As the regional $O_3$ transport reached the RL over the

35    western YRD, $O_3$ concentrations in the RL accumulated up to 200 µg m$^{-3}$ over the western site Nanjing in the sunrise hours of August 25. In accompany with the RL disappearance after sunrise, the vertical mixing initiated by convective and turbulent processes in the development of daytime convective boundary layer. The vertical mixing in the convective boundary layer after sunrise from the upper levels with entrainment of $O_3$-rich RL air to the ground with the net downward transport flux reaching up to 35 µg m$^{-3}$ h$^{-1}$, contributing a considerable surface $O_3$ accumulation to severe daytime $O_3$ pollution during summer smog

40    on August 25 in the western YRD region .  200  entrainment of $O_3$-rich RL air and boundary layer mixi~~ng contributed considerably to the rapid increase of surface $O_3$. Process analysis indicated vertical mixing contributed ~40µg m$^{-3}$ h$^{-1}$ of $O_3$ accumulation over Nanjing in the morning of August 25, 2016, which played an important role in contributing to the severe daytime $O_3$ pollution in the western YRD area.~~ The

45    mechanism of regional $O_3$ transport through the nocturnal RL revealed in this study has a great implication for understanding

O$_3$ pollution in air quality change.

**Key words:** Tropospheric O$_3$, residual layer, summer smog, regional O$_3$ transport, WRF-Chem

**1    Introduction**

Tropospheric ozone (O$_3$)O$_3$ is an important atmospheric composition influencing climate change and air quality in different
ways. According to the Intergovernmental Panel on Climate Change (IPCC, 2013), tropospheric O$_3$ is one of the most
important greenhouse gases for global warming. It is also a health hazard to sensitive individuals, reducing lung function and
contributing to exacerbation of asthma symptoms (Bell et al., 2006). Tropospheric O$_3$ is also important forcan alter
atmospheric chemistry because its photolysis in the presence of water vapor is the primary source for hydroxyl radical (OH),
which is responsible for the removal of many important trace gases. (Thompson, 1992; Logan et al., 1981).

The spatiotemporal variations of tropospheric O$_3$ are substantial in global and regional scales. In addition to photochemical
reactions in associated with O$_3$ precursor emissions and solar radiation, atmospheric transports of O$_3$ and its precursors play
an important role in determining the spatiotemporal distribution of tropospheric O$_3$, including horizontal transport (Wolff et al.,
1977; Yienger et al., 2000; Wild and Akimoto, 2001; Lelieveld et al., 2002; Duncan et al., 2008; Liu et al., 2011; Zhu et al.,
2017; Han et al., 2018) and vertical transport, e.g., exchange between stratosphere and troposphere (Hu et al., 2010; Jiang et al.,
2015) play an important role in determining the spatiotemporal distribution of tropospheric O$_3$.

Ambient O$_3$ levels are strongly influenced by diurnal variation of the atmospheric boundary layer (BL) structure. The daytime
BL, also known as the convective boundary layer (CBL), is directly affected by solar heating of the earth's surface. In the
major part of CBL, which is the mixing layer (ML), air pollutant concentrations distribute nearly uniformly resulted from the
convective turbulent mixing. The nocturnal BL is often characterized by a stable layer (SL) near the surface and an overlying
residual layer (RL). The SL develops due to radiative cooling after sunset. Above the SL, the remnants of the daytime ML form
the RL with initially uniformly mixed air pollutants remaining from the preceding daytime (Stull, 1988). (Stull, 1988), and O$_3$
is a representative remnant of air pollutants in the RL with the lack of O$_3$ consumption from NO titration and dry deposition
during nighttime (Xie et al., 2016; Sillman, 1999).

Nocturnal $O_3$ in the RL could exert an impact on the ambient $O_3$ variation during the following daytime (Aneja et al., 2000; Hu et al., 2012; Morris et al., 2010; Neu et al., 1994; Tong et al., 2011; Yorks et al., 2009; Hu et al., 2013; Klein et al., 2014). Locally, $O_3$ from the RL could contribute to the maximum surface $O_3$ on the following day with  the enhancement of surface $O_3$ by as much as 10-30 ppb (Hu et al., 2012). A few studies (Zhang et al., 1998; Zhang and Rao, 1999) investigated the  $O_3$ episodes in the BL over the northeastern United States (U.S.) based on measurements and 1-D model, suggesting that $O_3$ in the nocturnal RL could be transported to the downwind areas by the low-level jets over the eastern coast of U.S. Lee et al. (Lee et al., 2003) found that the daytime upslope flows transported $O_3$ precursors up to the mountain, while the nocturnal downslope flow brought the $O_3$-rich RL air mass downwards to Phoenix Valley, concluding that the transport, distribution and storage of $O_3$ are highly impacted by background meteorological conditions. Zhang et al. ( 2015) found that a regional transport within the RL from the surrounding urban areas could lead to a nighttime $O_3$ peak on a mountaintop in East China. However, the regional $O_3$ transport in the RL for air pollution has been poorly understood especially for plain areas.

In recent years, ambient $O_3$ levels have enhanced over the Yangtze River Delta (YRD) in East China with more frequent photochemical pollution events or summer smog from late May to July (Tang et al., 2013). During 1990 to 2013, the hourly $O_3$ peaks varied from 140 to 167 ppbv (about 294-350 μg m$^{-3}$) in the YRD region, from 160 to 180 ppbv (about 336-378 μg m$^{-3}$) in the Beijing-Tianjin-Hebei area over the North China Plain and from 200 to 220 ppbv (about 420-462 μg m$^{-3}$) in the Pearl River Delta of South China (Wang et al., 2017). Coupled with the increases of nitrogen oxides (NO$_x$) and volatile organic compounds (VOCs) emissions, $O_3$ distribution in the lower troposphere is significantly influenced by winds, air temperature, cloud cover, and downward shortwave radiation through changing the regional transport and chemical formation of $O_3$ (An et al., 2015; Gao et al., 2016; Xu et al., 2008; Li et al., 2018). $O_3$ levels could increase with a rate of 4-5 ppb K$^{-1}$ when temperature was between 28 and 38 ℃ (Pu et al., 2017). The prevailing winds driving transport of air pollutants from the YRD industrialized areas might have contributed to the $O_3$ enhancement (Tang et al., 2013).  Heat

wave with the maximum temperature ≥32 ℃ for 3 consecutive days  in the YRD with the sunny and  strong solar radiation environment  could significantly  strengthen photochemical reaction, potentially leading to substantial elevated $O_3$ in a warmer climate (Tie et al., 2009; Li et al., 2012; Wang et al., 2017; Xie et al., 2016; Pu et al., 2017).  Besides, the ambient $O_3$ level could be affected by diurnal variation of the atmospheric BL structure over the YRD with nighttime stable BL height dropping to 200 m and daytime BL height reaching up to about 1200 m (Chang et al., 2016). It is therefore important to understand the formation mechanisms of $O_3$ pollution including meteorological factors influencing $O_3$ pollution for summer smog over the YRD region.

This study focused on  the formation of $O_3$ pollution in a summer smog episode observed over the YRD in August 2016. We aimed to explore the underlying mechanism on regional $O_3$ transport over the YRD by using observational data and WRF-Chem modeling. The rest of this paper was organized as follows: section 2 described the observational data and the $O_3$ pollution episode. Section 3 presented the WRF-Chem model methodology and validation. In section 4, a mechanism of $O_3$ pollution formation was  revealed with regional $O_3$ transport in the RL from the eastern to western YRD. The conclusions were summarized in section 5.

**2    Observed $O_3$ pollution episode**

**2.1    Observation sites and data**

Observation data of the YRD urban sites of Nanjing (NJ), Zhenjiang (ZJ), Changzhou (CZ), Wuxi (WX), Suzhou (SZ) and Shanghai (SH) (Fig. 1) in August 2016 were used to study the $O_3$ pollution in summer smog episode. The  observation data of meteorology were collected from China Meteorological Administration (http://www.cnemc.cn) and the air pollutant  measurement chemical data from the national environmental monitoring network of China (http://www.mep.gov.cn). The meteorological data included wind speed (m s$^{-1}$) and direction (deg.) at 10 m above ground level, air temperature (℃) and relative humidity (%) at 2 m above ground level

with a temporal resolution of 3 hours, and total radiation irradiance with a time resolution of 1 hour. and. The air quality pollutant monitoring data The chemical data wer including O3 and NO2 e with a temporal resolution of 1 hour. The air pollutant concentrations were averaged from the measurements of multiple sites for each city, and the meteorological variables were only one site for each city.

**2.2 Summer smog in a heat wave episode over the YRD**

Due to With the w Western Pacific subtropical high staying over the YRD region in summer, heat wave events with it always leads to high surface air temperature and occur over this region in summer. heat wave. The averages of daily temperature and daily maximum temperature in YRD were 27.1 and 39.5 °C in summer during 2013-2016, and the average of summer daily temperature was 28.5 °C in NJ. It is generally accepted that high $O_3$ concentrations are accompanied by high air temperature with strong photochemical reactions (Filleul et al., 2006; Pu et al., 2017; Seinfeld and Pandis, 1986). During a the heat wave episode with the maximum temperature ≥32 °C for 3 consecutive days over August 22-25, 2016, a summer smog with severe $O_3$ pollution occurred over the YRD region (Table 1) and high surface $O_3$ concentrations with the averages of maximum 8-hour running mean values from 141.1 to 204.3 μg m$^{-3}$ were measured at all the 6 urban sites NJ, ZJ, CZ, WX, SZ and SH (Table 1). During this summer smog episode in mostly sunny days controlled by the westwards stretching subtropical anticyclone of the Western Pacific, the daily high maximum air temperature kept from 32.8 to 34.0 °C over the YRD region with mean air temperature over the periods of maximum 8-hour running mean surface $O_3$ concentrations air temperature averaged during the $O_3$ maximum 8-hours exceeding 32.0 °C. It is worthy note that t The $O_3$ concentrations over NJ, of the an urban site of the western YRD (site NJ), were 10-63 μg m$^{-3}$ much higher than those averaged over the eastern YRD region (sites CZ, WX, SZ and SH) during this heat wave episode. It is generally accepted that high $O_3$ concentrations are accompanied by high air temperature with strong photochemical reactions (Filleul et al., 2006; Pu et al., 2017; Seinfeld and Pandis, 1986).

**2.3 A potential role of regional $O_3$ transport**

Surface air temperature and solar radiation, deeply affect photochemical production. High levels of $O_3$ concentrations were

generally associated with high air temperatures (Rao et al., 1992; Council, 1991). The hourly maximum $O_3$ concentrations of site NJ were 256.8 and 317.2 µg m$^{-3}$ at 16:00 of August 24 and 15:00 of 25 (local time, same for hereinafter), showing a lag behind the time of maximum total radiation irradiances and maximum air temperature for a few hours (Fig. 2). However, we found from the observation of summer smog episode that the daily changes in air temperature and $O_3$ levels exhibited  the reversed  patterns in the western YRD area from August 24 to 25 (Fig. 2 and Table 2). The maximum 8-hour running mean $O_3$ concentrations and the maximum hourly $O_3$ concentrations respectively increased from 230.1 µg m$^{-3}$ and 256.8 µg m$^{-3}$ on August 24 to 284.8 µg m$^{-3}$ and 317.2 µg m$^{-3}$ on August 25 2016, presenting an obviously daily $O_3$ enhancement in  the western YRD area,  In contrast,  surface maximum total radiation irradiances and  maximum air temperature respectively decreased from 896 W m$^{-2}$ and 34.1 ℃ to 872 W m$^{-2}$ and 33.9 ℃  during the two days. This was a noteworthy observational evidence with the increasing surface $O_3$ concentrations in the ambient air with decreasing daytime air temperature and total radiation irradiance from August 24 to 25  in the western YRD (Figs. 1b-2, Table 2), which could be difficultly interpreted in the respect of photochemical production.

Both strong local photochemical production and atmospheric transport  lead to high surface $O_3$ concentrations (Jacob, 1999; Carnero et al., 2010; Corsmeier et al., 1997; Gangoiti et al., 2002; Godowitch et al., 2011; Tang et al., 2017; Shu et al., 2016). Based on the available observation of gaseous species, it is estimated that the  daytime mean surface nitrogen dioxide ($NO_2$) concentrations varied slightly during August 24 and 25 (Table 2), reflecting a less impact of the local photochemical production on the daily  enhancement of $O_3$ from August 24 to 25. The near-surface easterly winds prevailed in the directions of 90 deg. and 111 deg. with the daily averaged wind speeds of 2.4 and 2.6 m s$^{-1}$ respectively on August 24 and 25 at NJ (Table 2), indicating the fewer changes in both wind speed and direction over NJ during those two days. With excluding the impact of photochemical production and changes in horizontal winds, a potential role of regional $O_3$ transport  associated with vertical exchange over the YRD could become an important  mechnism in ambient $O_3$ pollution for the summer smog in the western YRD on August 25, 2016, which is explored with a modeling study in the following sections.

**3    Simulation settings and validation**

**3.1    Simulation settings**

To investigate the regional $O_3$ transport over the YRD and the underlying mechanism, the Weather Research and Forecasting model with Chemistry (WRF-Chem) version 3.8.1 is employed (Grell et al., 2005) in this study. Three nested domains respectively with the horizontal resolutions of 45, 15 and 5 km cover the areas of East Asia, East China and the YRD region (Fig. 1) with 32 vertical layers extending from the surface to 100 hPa. The simulation period spans from 21 to 30 25 August 2016 with 1-hourly model outputs and the spin-up time of first 24 hours. The physical parameterizations include Noah land-surface model (Tewari et al., 2004), Mesoscale Model (MM5) similarity surface layer, Yonsei University (YSU) boundary layer scheme (Hong et al., 2006), Rapid Radiative Transfer Model (RRTM) longwave scheme (Mlawer et al., 1997), Goddard shortwave scheme (Chou et al., 1998), Morrison double-moment microphysics scheme (Morrison et al., 2009), and Kain-Fritsch cumulus parameterization (Kain, 2004). The gas-phase chemical mechanism is selected with the Regional Acid Deposition Model, version 2 (RADM2) (Chang et al., 1990; Stockwell et al., 1984) including 158 chemical reactions among 36 species. The NCEP Final Global Forecast System Operational Analysis (FNL) data is used to provide the initial and boundary conditions of meteorological variables for the WRF-Chem simulation. The chemical initial and lateral boundary conditions are extracted from the global chemical transport Model for Ozone And Related chemical Tracers (MOZART)  model (Model for Ozone And Related chemical Tracers) (Emmons et al., 2010; Horowitz et al., 2003). The Multi-resolution Emission Inventory for China (MEIC) (http://www.meicmodel.org/) of year 2012 2016 is applied for the anthropocentric pollutant emissions, and the biogenic emissions are generated by the Model of Emissions of Gas and Aerosols from Nature (MEGAN) (Guenther et al., 2006).

**3.2    Modeling validation**

Simulated wind speed, air temperature, relative humidity, and $O_3$ concentrations are compared with the observationsThe simulations are compared with the wind speed, air temperature, relative humidity and $O_3$ concentrations observed at six sites over the YRD (Fig. 1b) during August 22-25, 2016 for the $O_3$ pollution episode (Fig. 3). The correlation coefficients for

near–surface air temperature and relative humidity reach up to  0.9 only with a slight overestimation of relative humidity. Over the sites NJ, CZ, WX, SZ and SH, the correlation coefficients between observed and simulated wind speed exceede 0.6. The high correlation coefficients between the observed and simulated $O_3$ concentrations are ranged between 0.8 and 0.9 with small standard deviations, and their normalized root-mean-square (NRMS) are  lower than 0.6. All the simulation and  observation correlations  passed the significant level of 0.001 (except wind speed over ZJ passing the significant level of 0.05). The validation of the vertical structures of $O_3$ is very important in the analysis of $O_3$ budget, but unavailable for us to evaluate the vertical structure of $O_3$ from simulation. If there would be observational data of $O_3$ vertical profiles, the validation of vertical profiles of $O_3$ could be done in future study of $O_3$ budget. In general, the WRF-Chem simulated $O_3$, air temperature, relative humidity and wind speed in the  YRD show a good agreement with the observations. The simulation reasonably captures the observed changes of $O_3$ and meteorology during the summer smog episode over the YRDTherefore, simulation data could be used to investigate the regional $O_3$ transport and the underlying mechanism over the YRD during the summer smog period, as presented in the following sections.

**4    Analysis  on regional $O_3$ transport**

**4.1   $O_3$ "reservoir" in the RL**

In order to analyze the development and evolution of $O_3$ "reservoir" in the residual layer (RL) during the summer smog, the time-altitude cross sections of $O_3$ concentrations and potential temperature over the western (site NJ) and eastern YRD (sites CZ, WX, SZ and SH) region are chosen to present the temporal changes in the vertical structures of $O_3$ concentrations and atmospheric boundary layer from August 24 to 25, 2016 based on the WRF-Chem  simulation (Fig. 4).

Figure 4a presents the hourly changes of vertical $O_3$ profiles from afternoon to midnight of August 24 over the eastern YRD region. In the afternoon of August 24, especially at 16:00 , the surface $O_3$ reached the peak concentrations of about 220 μg m$^{-3}$ in associated with the maximum air temperature during the heat wave, and the weak vertical gradients of potential temperature represented the well-developed mixing layer up to about 1.5 km height

above the surface for strong $O_3$ vertical mixing over the eastern YRD area (Fig. 4a). After the sunset on August 24, the near-surface $O_3$  decreased sharply with  ceasing photochemical production and the near-surface $O_3$ consumption  from nitrogen oxide (NO) titration and of dry deposition, forming a typical $O_3$-poor stable boundary layer and an overlying $O_3$ "reservoir" in the nocturnal RL over the eastern YRD region (Fig. 4a).

Figure 4b exhibits the hourly changes of vertical profiles of $O_3$ and potential temperature in the morning of August 25, 2016 over  the western YRD region. Reflected with the strong vertical gradients of potential temperature, the existence of the stable boundary layer up to 0.1 km height over the surface in the nighttime prevented $O_3$-rich air mass in the RL from vertical transport to the surface, building the $O_3$  "reservoir" in the RL in the altitudes from 0.1 km to 1 km  over the western YRD area (Fig 4b). After the sunrise on August 25, the stable boundary layer and RL vanished with the development of convective boundary layer (CBL), triggering  the vertical mixing of $O_3$-rich air mass in the RL and near-surface $O_3$-poor air mass in the  morning for redistributing the $O_3$ concentrations in the daytime CBL and enhancing the surface $O_3$ level (Fig. 4b).

We compared the temporal changes of $O_3$ "reservoir" in the nocturnal RL over the eastern  and western  YRD areas in Figures 4a and 4b, It is interesting that the eastern $O_3$ "reservoir" obviously leaked with reducing the $O_3$ concentrations over the nighttime of August 24 (Fig. 4a), while the western $O_3$ "reservoir" was gradually strengthened forming a high $O_3$ center exceeding 200 µg m$^{-3}$ over the site NJ around 6 am at the sunrise on August 25 (Fig. 4b). Considering the prevailing easterly winds in the lower troposphere over the YRD region during the summer smog period, we could speculate that the regional $O_3$ transport in the nocturnal RL could connect between the eastern decreases and western increases in the overnight $O_3$ "reservoir" within the RL over the YRD region (Figs. 4a and 4b). We further investigated  the regional $O_3$ transport in the nocturnal RL over the YRD to interpret the observational evidence of  exacerbating $O_3$ pollution in weakering photochemical production on August 25 in the  western YRD (Figs. 1b-2, Table 2).

**4.2  $O_3$ transport in the RL**

It is worth discussing why the nocturnal $O_3$ concentrations in the RL increased about 40 µg m$^{-3}$ over the western YRD region from 03:00 to 06:00 on August 25 (Fig. 4b). To investigate the regional $O_3$ transport over the YRD with contributing to the $O_3$ enhancement in the nocturnal RL over  the western YRD area, Figure 5 presents the variations of $O_3$ concentrations and wind streamlines at the altitude of about 900 m in the RL in the morning on 25 over the YRD. It is clearly seen from Figure 5 that the prevailing easterly winds drove the $O_3$ transport from the eastern to western YRD region during the nighttime from August 24 to 25, confirming our speculation about the regional $O_3$ transport in the nocturnal RL over the YRD connecting between the overnight changes in $O_3$ levels over the eastern and western areas (Figs. 4a and 4b). It is noteworthy that the regional $O_3$ transport in the nocturnal RL reached westwards over the western urban site NJ  before the sunrise around 6:00 on August 25, in associated with the stagnation of cyclone circulation over NJ from 6:00 until 10:00, which prevented high $O_3$ from moving further west and converged $O_3$ into the RL over the western YRD region (sites  NJ and ZJ) until the sunrise around 6:00 on August 25 ,2016 (Fig. 5).

Figure 6 presents the temporal evolution of vertical sections of $O_3$ concentrations and atmospheric circulation along the regional $O_3$ transport over the YRD region to further explore the mechanism of regional $O_3$ transport over the YRD for summer smog. The vertical distributions of $O_3$ concentrations were controlled strongly by the diurnal change of BL structure. The daytime $O_3$ concentrations distributed vertically uniformly in the mixing layer (ML), the major part of CBL over the YRD (Fig. 6a), which could form the $O_3$-rich RL after the sunset.  Under the guidance of the prevailing easterly winds, the $O_3$ transport from the eastern to western YRD region persisted during the nighttime from August 24 to 25 (Figs. 6b-6e), confirming our speculation about the regional $O_3$ transport in the nocturnal RL over the YRD region. The regional $O_3$ transport in the RL from the eastern to western YRD site NJ enhanced the $O_3$ concentrations up to 200 µg m$^{-3}$ within the RL over western site NJ during the sunrise hours of August 25 (Fig 4b). The $O_3$ horizontal transport flux in RL averaged over the nighttime from 20:00 on August 24 to 8:00 on 25 was 541 µg m$^{-2}$ s$^{-1}$ at the western site NJ with 119 µg m$^{-2}$ s$^{-1}$ stronger than that during the preceding night to August 24, reflecting the larger contribution of $O_3$ horizontal transport in RL to the $O_3$ pollution on August 25 over the western YRD.the western site NJ, The RL with $O_3$-rich air mass over the

western area , contributed by the nocturnal $O_3$ transport over the YRD, was broken with  development of the daytime CBL with strong vertical mixing after the sunrise on August 25 (Fig. 6f). The daily large contribution of vertical mixing to the surface $O_3$ level occurred around 10:00 in the morning with downwards vertical mixing from the upper levels with entrainment of $O_3$-rich RL air to the ground $O_3$-poor air  for summer smog (Figs. 6f and 7).

**4.3 Contribution of $O_3$ vertical mixing from the RL**

As discussed in sections 4.1 and 4.2, $O_3$-rich air mass could transport from east to west in nocturnal RL over the YRD. The CBL  establishes after the sunrise during daytime. Consequently the $O_3$-rich air mass could be entrained downwards to the surface (Mcelroy and Smith, 1993; Venkatram, 1977), contributing to the surface $O_3$ concentrations early in the day time (Fig. 6f).

Based on the WRF-Chem simulation, Figure 7 presents the hourly changes in the contribution rates of vertical mixing and local chemical reactions to surface $O_3$ in the western urban site NJ  over August 24-25. Vertical mixing initiated by convective and turbulent processes in the development of daytime convective boundary layer , and chemical reactions are net output of all $O_3$ chemical reactions (Gao et al., 2016). The positive and negative contribution rates indicate respectively the gain and loss of surface $O_3$ concentrations through vertical mixing and local chemical reactions. The daily totals of positive contribution of vertical mixing and local chemical reactions on August 24 and 25 are given in Table 3. Relatively to August 24, the positive contribution of $O_3$ vertical mixing enhanced significantly on August 25 with the largest contribution of about  35 µg m$^{-3}$ h$^{-1}$, twice as that on August 24 (Fig. 7). Tropospheric $O_3$ results mainly from photochemical reactions in daytime (Seinfeld and Pandis, 1986). Although the largest contributions of chemical reactions reached up to  38 µg m$^{-3}$ h$^{-1}$ and  44 µg m$^{-3}$ h$^{-1}$ in the afternoon respectively on August 24 and 25 (Fig. 7), the daily totals of positive contributions of chemical reactions were estimated to be lower on August 25 with  238 µg m$^{-3}$ than the previous daytime of August 24 with  240 µg m$^{-3}$ in the western YRD area. The daily totals of positive $O_3$ contribution of vertical mixing raised sharply to  115 µg m$^{-3}$ on August 25 with a large increase of  52 µg m$^{-3}$ from August 24 (Table 3).

The high $O_3$ levels in ambient air for summer smog in the western YRD on August 25 were significantly contributed from the downwards vertical mixing of $O_3$-rich RL air mass, which was transported in the nocturnal previous nighttime RL from the eastern to western YRD. The regional $O_3$ transport in the nocturnal RL in associated with the diurnal changes of boundary layer are revealed to be an important mechanism of regional $O_3$ transport in East China.

Based on the simulated dry depositions, we calculate the hourly changes of $O_3$ dry depositions and estimated the daily averages of dry deposition rates with about 0.42 and 0.49 μg m$^{-2}$ s$^{-1}$ respectively for August 24 and 25. The dry depositions of $O_3$ varied little over these two days with a slight enhancement on August 25, reflecting $O_3$ dry depositions exerted less impact on surface $O_3$ change during August 24-25. The contribution of $O_3$ dry deposition to tropospheric $O_3$ changes was trivial compared to vertical mixing and chemical reactions (Wang et al., 1998; Fowler et al., 1999; Zaveri et al., 2003).

Considering weak changes of local emissions in short time, the WRF-Chem simulation with the hourly emissions of chemical species over YRD unchanged between August 24 and 25. To analyze the impact from photochemical production, we used the surface $NO_2$ concentrations and total radiation irradiance (TRI) to analyze the change of photochemical production rates. There were no apparent changes of $NO_2$ and TRI between August 24 and 25, indicating that photochemical production exerted less impact on the high $O_3$ level on August 25 compared to regional $O_3$ transport in nocturnal RL. The analysis of simulation results, revealed that vertical mixing from the upper $O_3$-rich RL to daytime surface layer was a large contributor to $O_3$ enhancement for summer smog in the western YRD on August 25, 2016 (Fig. 7).

**5    Conclusions**

By analyzing the observational data of gaseous species and meteorological variables during severe summer smog over the YRD in East China in August, 2016, we found a noteworthy observational evidence with of the increased increasing daytime surface $O_3$ concentrations levels with excluding the impact of photochemical production in the ambient air of lower daytime air temperature and weaker solar radiation from August 24 to 25 in the western YRD with excluding the impact of

photochemical production. Regional $O_3$ transport over the YRD could play  a more important role in the ambient $O_3$ pollution on August 25 in the western YRD.

By combining environmental and meteorological observation data with air quality modeling, the formation mechanism of $O_3$ pollution episode over the YRD area, particularly the severe pollution over western YRD was investigated. On August 24, the high $O_3$ levels peaked at about  220 μg m$^{-3}$ in the daytime mixing layer over the eastern YRD area. During nighttime, a shallow stable boundary layer formed near the surface, decoupled the RL above it with an $O_3$-rich "reservoir". Governed by prevailing easterly wind in the lower troposphere, the $O_3$-rich air mass in the nocturnal RL shifted from the eastern to western YRD with the horizontal transport flux of 541 μg m$^{-2}$ s$^{-1}$. Consequently, the $O_3$ concentrations in the RL over the western YRD area enhanced up to  200 μg m$^{-3}$  around sunrise of August 25, 2016. In accompany with the disappearance of the RL after sunrise, the vertical mixing initiated by convective and turbulent processes in the establishment of daytime CBL. The vertical mixing in the CBL from the upper levels to the ground with the net downward transport flux reaching up to 35 μg m$^{-3}$ h$^{-1}$ during daytime on August 25, contributing a considerable surface $O_3$ accumulation to summer smog over the western YRD region on ~~In accompany with the growth of the convective boundary layer breaking up the RL after the sunrise, entrainment of $O_3$-rich RL air and boundary layer mixing contributed considerably to the rapid increase of surface $O_3$. Process analysis indicated vertical mixing contributed ~40μg m$^{-3}$ h$^{-1}$ of $O_3$ accumulation over the western YRD in the morning ofwasdaytime~~ $O_3$ pollution in the western YRD.

This study  revealed an important mechanism of regional $O_3$ transport through the nocturnal RL from upstream to downstream areas driven by the prevailing winds in the lower troposphere in closely associated with the diurnal change  in the atmospheric boundary layer, which could be depicted with a conceptual model in Figure 8. This mechanism of regional $O_3$ transport has a substantial implication for understanding urban $O_3$ pollution in air quality change.

The regional $O_3$ transport in atmospheric boundary layer in this case of summer smog in the YRD, East China is to be further studied with more comprehensive observations of meteorology and environment including impact of changes in biogenic VOC on $O_3$ concentrations, as well as the better modeling of  atmospheric boundary layer.

330    **Acknowledgements**

This study was jointly funded by National Key R & D Program of China (2016YFC0203304), National Natural Science Foundation of China (91744209; 91644223), the National Key Basic Research Development Program of China (2014CB441203), Program of Shanghai's committee of Science and Technology (16DZ1204607), and Postgraduate Research & Practice Innovation Program of Jiangsu Province (KYCX18_1003).

335    **References**

An, J., Zou, J., Wang, J., Lin, X., and Zhu, B.: Differences in ozone photochemical characteristics between the megacity Nanjing and its suburban surroundings, Yangtze River Delta, China, Environmental Science and Pollution Research, 22, 19607, 2015.

Aneja, V. P., Mathur, R., Arya, S. P., Li, Y., Murray, G. C., and Manuszak, T. L.: Coupling the Vertical Distribution of Ozone
340    in the Atmospheric Boundary Layer, Environmental Science & Technology, 34, 15-28, 2000.

Bell, M. L., Peng, R. D., and Dominici, F.: The Exposure-Response Curve for Ozone and Risk of Mortality and the Adequacy of Current Ozone Regulations, Environmental Health Perspectives, 114, 532-536, 2006.

Carnero, J. A., Bolívar, J. P., and Ba, D. L. M.: Surface ozone measurements in the southwest of the Iberian Peninsula (Huelva, Spain), Environmental Science and Pollution Research, 17, 355-368, 2010.

345    Chang, J. S., Middleton, P. B., Stockwell, W. R., Binkowski, F. S., and Byun, D.: Acidic deposition: State of science and technology. Report 4. The regional acid deposition model and engineering model. Final report, Journal of Materials Science Letters, 9, 772-773, 1990.

Chang, Y., Zou, Z., Deng, C., Huang, K., Collett, J. L., Lin, J., and Zhuang, G.: The importance of vehicle emissions as a source of atmospheric ammonia in the megacity of Shanghai, Atmospheric Chemistry and Physics, 16, 3577, 2016.

350    Chou, M. D., Suarez, M. J., Ho, C. H., Yan, M. M. H., and Lee, K. T.: Parameterizations for Cloud Overlapping and Shortwave Single-Scattering Properties for Use in General Circulation and Cloud Ensemble Models, Journal of Climate, 11, 202-214, 1998.

Corsmeier, U., Kalthoff, N., Kolle, O., Kotzian, M., and Fiedler, F.: Ozone concentration jump in the stable nocturnal boundary layer during a LLJ-event, Atmospheric Environment, 31, 1977-1989, 1997.

355    Council, N. R.: Rethinking the ozone problem in urban and regional air pollution, National Academy Press, 1991.

Duncan, B. N., West, J. J., Yoshida, Y., Fiore, A. M., and Ziemke, J. R.: The influence of European pollution on ozone in the Near East and northern Africa, Atmospheric Chemistry & Physics, 8, 2267-2283, 2008.

Emmons, L. K., Walters, S., Hess, P. G., Lamarque, J. F., Pfister, G. G., Fillmore, D., Granier, C., Guenther, A., Kinnison, D., and Laepple, T.: Description and evaluation of the model for ozone and related chemical tracers, version 4 (MOZART-4),
360    Geoscientific Model Development, 3, 43-67, 2010.

Filleul, L., Cassadou, S., Médina, S., Fabres, P., Lefranc, A., Eilstein, D., Tertre, A. L., Pascal, L., Chardon, B., and Blanchard, M.: The Relation between Temperature, Ozone, and Mortality in Nine French Cities during the Heat Wave of 2003, Environmental Health Perspectives, 114, 1344, 2006.

Fowler, D., Cape, J., Coyle, M., Smith, R., Hjellbrekke, A.-G., Simpson, D., Derwent, R., and Johnson, C.: Modelling photochemical oxidant formation, transport, deposition and exposure of terrestrial ecosystems, Environmental Pollution, 100, 43-55, 1999.

Gangoiti, G., Alonso, L., Navazo, M., Albizuri, A., Perez-Landa, G., Matabuena, M., Valdenebro, V., Maruri, M., GarcíA, J. A., and Millán, M. M.: Regional transport of pollutants over the Bay of Biscay: analysis of an ozone episode under a blocking anticyclone in west-central Europe, Atmospheric Environment, 36, 1349-1361, 2002.

Gao, J., Zhu, B., Xiao, H., Kang, H., Hou, X., and Shao, P.: A case study of surface ozone source apportionment during a high concentration episode, under frequent shifting wind conditions over the Yangtze River Delta, China, Science of the Total Environment, 544, 853, 2016.

Godowitch, J. M., Gilliam, R. C., and Rao, S. T.: Diagnostic evaluation of ozone production and horizontal transport in a regional photochemical air quality modeling system, Atmospheric Environment, 45, 3977-3987, 2011.

Grell, G. A., Peckham, S. E., Schmitz, R., McKeen, S. A., Frost, G., Skamarock, W. C., and Eder, B.: Fully coupled "online" chemistry within the WRF model, Atmospheric Environment, 39, 6957-6975, 2005.

Guenther, A., Karl, T., Harley, P., Wiedinmyer, C., Palmer, P. I., and Geron, C.: Estimates of global terrestrial isoprene emissions using MEGAN (Model of Emissions of Gases and Aerosols from Nature), Atmospheric Chemistry & Physics, 6, 3181-3210, 2006.

Han, H., Liu, J., Yuan, H., Zhuang, B., Zhu, Y., Wu, Y., Yan, Y., and Ding, A.: Characteristics of intercontinental transport of tropospheric ozone from Africa to Asia, Atmos. Chem. Phys., 18, 4251-4276, 10.5194/acp-18-4251-2018, 2018.

Hong, S.-Y., Noh, Y., and Dudhia, J.: A New Vertical Diffusion Package with an Explicit Treatment of Entrainment Processes, Monthly Weather Review, 134, 2318-2341, 10.1175/mwr3199.1, 2006.

Horowitz, L. W., Walters, S., Mauzerall, D. L., Emmons, L. K., Rasch, P. J., Granier, C., Tie, X., Lamarque, J. F., Schultz, M. G., and Tyndall, G. S.: A global simulation of tropospheric ozone and related tracers: Description and evaluation of MOZART, version 2, Journal of Geophysical Research Atmospheres, 108, ACH 16-11, 2003.

Hu, X.-M., Fuentes, J. D., and Zhang, F.: Downward transport and modification of tropospheric ozone through moist convection, Journal of atmospheric chemistry, 65, 13-35, 2010.

Hu, X.-M., Klein, P. M., Xue, M., Zhang, F., Doughty, D. C., Forkel, R., Joseph, E., and Fuentes, J. D.: Impact of the vertical mixing induced by low-level jets on boundary layer ozone concentration, Atmospheric Environment, 70, 123-130, 2013.

Hu, X. M., Doughty, D. C., Sanchez, K. J., Joseph, E., and Fuentes, J. D.: Ozone variability in the atmospheric boundary layer in Maryland and its implications for vertical transport model, Atmospheric Environment, 46, 354-364, 2012.

Jacob, D.: Introduction to atmospheric chemistry, Princeton University Press, 1999.

Jiang, Y., Zhao, T., Liu, J., Xu, X., Tan, C., Cheng, X., Bi, X., Gan, J., You, J., and Zhao, S.: Why does surface ozone peak before a typhoon landing in southeast China?, Atmospheric Chemistry and Physics, 15, 13331-13338, 2015.

Kain, J. S.: The Kain–Fritsch Convective Parameterization: An Update, Journal of Applied Meteorology, 43, 170-181, 10.1175/1520-0450(2004)043<0170:tkcpau>2.0.co;2, 2004.

Klein, P. M., Hu, X.-M., and Xue, M.: Impacts of mixing processes in nocturnal atmospheric boundary layer on urban ozone concentrations, Boundary-layer meteorology, 150, 107-130, 2014.

400 Lee, S. M., Fernando, H. J. S., Princevac, M., Zajic, D., Sinesi, M., Mcculley, J. L., and Anderson, J.: Transport and diffusion of ozone in the nocturnal and morning planetary boundary layer of the Phoenix Valley, Environmental Fluid Mechanics, 3, 331-362, 2003.

Lelieveld, J., Berresheim, H., Borrmann, S., Crutzen, P. J., Dentener, F. J., Fischer, H., Feichter, J., Flatau, P. J., Heland, J., and Holzinger, R.: Global air pollution crossroads over the Mediterranean, Science, 298, 794-799, 2002.

405 Li, L., Chen, C., Huang, C., Huang, H., Zhang, G., Wang, Y., Wang, H., Lou, S., Qiao, L., and Zhou, M.: Process analysis of regional ozone formation over the Yangtze River Delta, China using the Community Multi-scale Air Quality modeling system, Atmospheric Chemistry and Physics, 12, 10971-10987, 2012.

Li, S., Wang, T., Huang, X., Pu, X., Li, M., Chen, P., Yang, X. Q., and Wang, M.: Impact of East Asian summer monsoon on surface ozone pattern in China, Journal of Geophysical Research: Atmospheres, 123, 1401-1411, 2018.

410 Liu, J. J., Jones, D. B. A., Zhang, S., and Kar, J.: Influence of interannual variations in transport on summertime abundances of ozone over the Middle East, Journal of Geophysical Research Atmospheres, 116, 999-1010, 2011.

Logan, J. A., Prather, M. J., Wofsy, S. C., and Mcelroy, M. B.: Tropospheric chemistry: A global perspective, Journal of Geophysical Research Oceans, 86, 7210-7254, 1981.

Mcelroy, J. L., and Smith, T. B.: Creation and fate of ozone layers aloft in Southern California, Atmospheric
415 Environment.part A.general Topics, 27, 1917-1929, 1993.

Mlawer, E. J., Taubman, S. J., Brown, P. D., Iacono, M. J., and Clough, S. A.: Radiative transfer for inhomogeneous atmospheres: RRTM, a validated correlated-k model for the longwave, Journal of Geophysical Research: Atmospheres, 102, 16663-16682, 10.1029/97JD00237, 1997.

Morris, G. A., Ford, B., Rappenglück, B., Thompson, A. M., Mefferd, A., Ngan, F., and Lefer, B.: An evaluation of the
420 interaction of morning residual layer and afternoon mixed layer ozone in Houston using ozonesonde data, Atmospheric Environment, 44, 4024-4034, 2010.

Morrison, H., Thompson, G., and Tatarskii, V.: Impact of Cloud Microphysics on the Development of Trailing Stratiform Precipitation in a Simulated Squall Line: Comparison of One- and Two-Moment Schemes, Monthly Weather Review, 137, 991-1007, 10.1175/2008mwr2556.1, 2009.

425 Neu, U., Künzle, T., and Wanner, H.: On the relation between ozone storage in the residual layer and daily variation in near-surface ozone concentration — A case study, Boundary-Layer Meteorology, 69, 221-247, 1994.

Pu, X., Wang, T., Huang, X., Melas, D., Zanis, P., Papanastasiou, D., and Poupkou, A.: Enhanced surface ozone during the heat wave of 2013 in Yangtze River Delta region, China, Science of the Total Environment, 603, 807-816, 2017.

Rao, S. T., Sistla, G., and Henry, R.: Statistical analysis of trends in urban ozone air quality, Journal of the Air & Waste
430 Management Association, 42, 1204-1211, 1992.

Seinfeld, J. H., and Pandis, S. N.: Atmospheric chemistry and physics: From air pollution to climate change (Second Edition), Wiley, 1595-1595 pp., 1986.

Sillman, S., The relation between ozone, NOx and hydrocarbons in urban and polluted rural environments. Atmos. Environ. 33, 1821–1845, 1999.

435    Shu, L., Xie, M., Wang, T., Gao, D., Chen, P., Han, Y., Li, S., Zhuang, B., and Li, M.: Integrated studies of a regional ozone pollution synthetically affected by subtropical high and typhoon system in the Yangtze River Delta region, China, Atmospheric Chemistry and Physics, 16, 15801-15819, 2016.

Stockwell, W. R., Middleton, P., Chang, J. S., and Tang, X.: The second generation regional acid deposition model chemical mechanism for regional air quality modeling, 190–197 pp., 1984.

440    Stull, R. B.: An introduction to boundary layer meteorology, Atmospheric Sciences Library, 8, 89, 1988.

Tang, G., Zhu, X., Xin, J., Hu, B., Song, T., Sun, Y., Zhang, J., Wang, L., Cheng, M., and Chao, N.: Modelling study of boundary-layer ozone over northern China-Part I: Ozone budget in summer, Atmospheric Research, 187, 128-137, 2017.

Tang, H., Liu, G., Zhu, J., Han, Y., and Kobayashi, K.: Seasonal variations in surface ozone as influenced by Asian summer monsoon and biomass burning in agricultural fields of the northern Yangtze River Delta, Atmospheric research, 122, 67-76,

445    2013.

Taylor, K. E.: Summarizing multiple aspects of model performance in a single diagram, Journal of Geophysical Research: Atmospheres, 106, 7183-7192, doi:10.1029/2000JD900719, 2001.

Tewari, M., Chen, F., Wang, W., Dudhia, J., Lemone, M. A., Mitchell, K., Ek, M., Gayno, G., Wegiel, J., and Cuenca, R. H.: Implementation and verification of the unified NOAH land surface model in the WRF model, Conference on Weather

450    Analysis and Forecasting/ Conference on Numerical Weather Prediction, 2004, 11-15.

Thompson, A. M.: The oxidizing capacity of the earth's atmosphere: probable past and future changes, Science, 256, 1157-1165, 1992.

Tie, X., Geng, F., Peng, L., Gao, W., and Zhao, C.: Measurement and modeling of O3 variability in Shanghai, China: Application of the WRF-Chem model, Atmospheric Environment, 43, 4289-4302, 2009.

455    Tong, N. Y. O., Leung, D. Y. C., and Liu, C. H.: A Review on Ozone Evolution and Its Relationship with Boundary Layer Characteristics in Urban Environments, Water, Air, & Soil Pollution, 214, 13-36, 2011.

Venkatram, A.: Internal boundary layer development and fumigation, Atmospheric Environment, 11, 479-482, 1977.

Wang, T., Xue, L., Brimblecombe, P., Lam, Y. F., Li, L., and Zhang, L.: Ozone pollution in China: A review of concentrations, meteorological influences, chemical precursors, and effects, Science of the Total Environment, 575, 1582-1596, 2017.

460    Wang, Y., Logan, J. A., and Jacob, D. J.: Global simulation of tropospheric O3-NOx-hydrocarbon chemistry: 2. Model evaluation and global ozone budget, Journal of Geophysical Research Atmospheres, 103, 10713-10725, 1998.

Wild, O., and Akimoto, H.: Intercontinental transport of ozone and its precursors in a three‐dimensional global CTM, Journal of Geophysical Research Atmospheres, 106, 27729-27744, 2001.

Wolff, G. T., Lioy, P. J., Wight, G. D., Meyers, R. E., and Cederwall, R. T.: An investigation of long-range transport of ozone

465    across the midwestern and eastern united states, Atmospheric Environment, 11, 797-802, 1977.

Xie, M., Zhu, K., Wang, T., Chen, P., Han, Y., Li, S., Zhuang, B., and Shu, L.: Temporal characterization and regional contribution to O3 and NOx at an urban and a suburban site in Nanjing, China, Science of the Total Environment, 551, 533-545, 2016.

Xu, X., Lin, W., Wang, T., Yan, P., Tang, J., Meng, Z., and Wang, Y.: Long-term trend of surface ozone at a regional

470    background station in eastern China 1991–2006: enhanced variability, Atmospheric Chemistry and Physics, 8, 2595-2607, 2008.

Yienger, J. J., Galanter, M., Holloway, T. A., Phadnis, M. J., Guttikunda, S. K., Carmichael, G. R., Moxim, W. J., and Hiram Levy, I. I.: The Episodic Nature of Air Pollution Transport From Asia to North America, Journal of Geophysical Research Atmospheres, 105, 26931-26946, 2000.

475 Yorks, J. E., Thompson, A. M., Joseph, E., and Miller, S. K.: The variability of free tropospheric ozone over Beltsville, Maryland (39N, 77W) in the summers 2004–2007, Atmospheric Environment, 43, 1827-1838, 2009.

Zaveri, R. A., Berkowitz, C. M., Kleinman, L. I., Springston, S. R., Doskey, P. V., Lonneman, W. A., and Spicer, C. W.: Ozone production efficiency and NOx depletion in an urban plume: Interpretation of field observations and implications for evaluating O3-NOx-VOC sensitivity, Journal of Geophysical Research: Atmospheres, 108, 2003.

480 Zhang, J., Rao, S. T., and Daggupaty, S. M.: Meteorological processes and ozone exceedances in the northeastern United States during the 12-16 July 1995 Episode*, Journal of Applied Meteorology, 37, 776-789, 1998.

Zhang, J., and Rao, S. T.: The role of vertical mixing in the temporal evolution of ground-level ozone concentrations, Journal of Applied Meteorology, 38, 1674-1691, 1999.

Zhang, L., Jin, L., Zhao, T., Yan, Y., Zhu, B., Shan, Y., Guo, X., Tan, C., Gao, J., and Wang, H.: Diurnal variation of surface
485 ozone in mountainous areas: Case study of Mt. Huang, East China, Science of the Total Environment, 538, 583-590, 2015.

Zhu, Y., Liu, J., Wang, T., Zhuang, B., Han, H., Wang, H., Chang, Y., and Ding, K.: The Impacts of Meteorology on the Seasonal and Interannual Variabilities of Ozone Transport from North America to East Asia, Journal of Geophysical Research Atmospheres, 122, 2017.

490

**Table 1: Averages (Ave) of maximum 8-hour running mean surface $O_3$ concentrations (μg m$^{-3}$), mean air temperature (℃) over the periods of maximum 8-hour running mean surface $O_3$ concentrations and daily maximum air temperature (℃) with their standard deviations (Std) over August 22-25, 2016 observed at site NJ in the western YRD.**

| Sites | Maximum 8-hour running mean $O_3$ | | Mean air temperature | | Daily maximum air temperature | |
|---|---|---|---|---|---|---|
| | Ave | Std | Ave | Std | Ave | Std |
| NJ | 204.3 | 58.2 | 32.6 | 0.5 | 33.4 | 0.7 |
| ZJ | 163.3 | 44.7 | 32.6 | 0.7 | 33.2 | 0.8 |
| CZ | 174.4 | 34.8 | 33.2 | 0.7 | 34.2 | 0.9 |
| WX | 190.5 | 24.9 | 33.4 | 0.3 | 34.5 | 0.6 |
| SZ | 173.7 | 21.0 | 33.3 | 0.2 | 34.0 | 0.2 |
| SH | 141.1 | 7.8 | 32.0 | 0.4 | 32.7 | 0.3 |

495 **Table 2: Meteorological and environmental elements observed at site NJ in the western YRD from August 24 to 25, 2016 with their daily differences ($\Delta$x).**

| | Aug. 24 | Aug. 25 | $\Delta$x |
|---|---|---|---|
| Maximum 8-hour running mean surface $O_3$ concentrations ($\mu g\ m^{-3}$) | 230.1 | 284.8 | 54.7 |
| Maximum hourly surface $O_3$ concentration ($\mu g\ m^{-3}$) | 256.8 | 317.2 | 60.4 |
| Daytime mean surface $O_3$ concentrations ($\mu g\ m^{-3}$) | 180.6 | 230.1 | 49.5 |
| Daytime mean surface $NO_2$ concentrations ($\mu g\ m^{-3}$) | 27.9 | 27.8 | - 0.1 |
| Daily maximum air temperature at 2 m (℃) | 34.1 | 33.9 | - 0.2 |
| Maximum surface total radiation irradiance (W $m^{-2}$) | 896.0 | 872.0 | - 24.0 |
| Daytime mean surface total radiation irradiance (W $m^{-2}$) | 511.8 | 423.4 | - 88.4 |
| Daily mean wind speed at 10 m (m $s^{-1}$) | 2.4 | 2.6 | 0.2 |
| Daily mean wind direction at 10 m (deg.) | 90 | 111 | 21 |

**Table 3: Comparisons in the daily totals of positive contribution (µg m$^{-3}$) of vertical mixing (VMIX) and chemical reactions (CHEM) to surface O$_3$ concentrations in the western YRD area between August 24 and 25, 2016.**

| Date | VMIX | CHEM |
|------|------|------|
| Aug. 24 | 63 | 240 |
| Aug. 25 | 115 | 238 |

500

[Figure]

**Figure 1: (a) Three nesting domains for the WRF-Chem simulation, (b) the topography and the locations of 6 observation sites over the YRD region in the fine domain d03.**

[Figure]

505

**Figure 2: Time series of surface O$_3$ concentrations (O$_3$), 2 m air temperature (Temp) and surface total radiation irradiance (TRI) observed at site NJ.**

[Figure]

**Figure 3: Modeling validations in the Taylor plots with the standard deviations and correlation coefficients of simulated and observed meteorological elements and surface O$_3$ concentrations at six YRD sites (Fig. 1). The azimuthal angle represents correlation coefficient, the radial distance the ratio of standard deviation between simulations and observations, and the semicircles centered at the "REF" marker the normalized root-mean-square (NRMS) (Taylor, 2001).**

[Figure]

**Figure 4: Time-altitude cross sections of O$_3$ concentrations and potential temperature over (a) the eastern YRD area covering the**

515    **sites CZ, WX, SZ and SH from at 14:00 of August 24 to 0:00 on August 25 and (b) the western YRD site NJ from 00:00 to 12:00 on**

    **August 25.**

[Figure]

**Figure 5: The spatial distribution of O₃ concentrations and wind streamlines at about 900 m height at (a) 00:00, (b) 03:00, (c) 06:00,**

**(d) 09:00 on August 25, 2016. The blue thick lines with arrows represent the major routes of regional O$_3$ transport from the eastern**

 **to western YRD.**

[Figure]

**Figure 6: Vertical sections of O$_3$ concentrations (contours) and atmospheric circulation (wind vectors) along the regional O$_3$ transport route (red solid line in Fig. 1b) from east to west over the YRD region with the box columns W and E respectively marking the western YRD area surrounding NJ and the eastern YRD area covering CZ, WX, SZ and SH from August 24 to 25,**

 **2016. The vertical wind velocities are multiplied by 50 for illustration of vertical circulations.**

[Figure]

**Figure 7: The contribution rates of vertical mixing (VMIX) and chemical reactions (CHEM) to surface O₃ concentrations at site NJ during August 24 -25, 2016.**

[Figure]

530       **Figure 8: A diagram of regional O$_3$ transport mechanism proposed in this study.**